# The transcriptional regulator ZNF398 mediates pluripotency and epithelial character downstream of TGF-beta in human PSCs

Irene Zorzan [1,8], Marco Pellegrini [1,5,8], Mattia Arboit[1], Danny Incarnato[2,3,6], Mara Maldotti[2,3], Mattia Forcato[4], Guidantonio Malagoli Tagliazucchi[4,7], Elena Carbognin [1], Marco Montagner [1], Salvatore Oliviero [2,3✉] & Graziano Martello [1✉]

Human pluripotent stem cells (hPSCs) have the capacity to give rise to all differentiated cells of the adult. TGF-beta is used routinely for expansion of conventional hPSCs as flat epithelial colonies expressing the transcription factors POU5F1/OCT4, NANOG, SOX2. Here we report a global analysis of the transcriptional programme controlled by TGF-beta followed by an unbiased gain-of-function screening in multiple hPSC lines to identify factors mediating TGF-beta activity. We identify a quartet of transcriptional regulators promoting hPSC self-renewal including ZNF398, a human-specific mediator of pluripotency and epithelial character in hPSCs. Mechanistically, ZNF398 binds active promoters and enhancers together with SMAD3 and the histone acetyltransferase EP300, enabling transcription of TGF-beta targets. In the context of somatic cell reprogramming, inhibition of ZNF398 abolishes activation of pluripotency and epithelial genes and colony formation. Our findings have clear implications for the generation of bona fide hPSCs for regenerative medicine.

[1] Department of Molecular Medicine, Medical School, University of Padua, 35121 Padua, Italy. [2] Department of Life Sciences and Systems Biology and Molecular Biotechnology Center (MCB), University of Turin, 10126 Turin, Italy. [3] Italian Institute for Genomic Medicine (IIGM), 10060 Candiolo (TO), Italy. [4] Department of Life Sciences, University of Modena and Reggio Emilia, 41125 Modena, Italy. [5] Present address: UCL Great Ormond Street Institute of Child Health, Developmental Biology and Cancer, Stem Cells and Regenerative Medicine, 30 Guilford Street, WC1N 1EH London, UK. [6] Present address: Department of Molecular Genetics, Groningen Biomolecular Sciences and Biotechnology Institute (GBB), University of Groningen, Nijenborgh 7, 9747 AG Groningen, the Netherlands. [7] Present address: UCL Genetics Institute, Department of Genetics, Evolution and Environment, University College London, Darwin Building, WC1E 6BT London, UK. [8] These authors contributed equally: Irene Zorzan, Marco Pellegrini. ✉email: salvatore.oliviero@unito.it; graziano.martello@unipd.it

Human pluripotent stem cells (hPSCs) have been derived from human blastocysts as human embryonic stem cells (hESCs[1]) or from somatic cells via transcription factor-mediated reprogramming as induced pluripotent stem cells (hiPSCs[2,3]). Initially hPSCs were cultured on layers of inactivated fibroblasts (feeder cells), which produce several adhesion and signalling molecules. Alternatively, the medium was conditioned, or enriched by unknown secreted factors, by fibroblasts before their use with hPSCs. Such poorly defined culture systems represented a hurdle to the identification of key signals regulating pluripotency. Importantly, chemically defined conditions for the expansion of hPSCs have been reported[4–7]. Despite variations in the media composition, ligands of the TGF-beta family are invariably added or produced by feeder cells[8,9]. Indeed, TGF-beta signalling has been shown to be critical for the maintenance of pluripotency in hPSCs[10,11]. However, the mechanisms of action of the TGF-beta signal remain poorly characterised.

TGF-beta ligands such as TGF-beta1/2/3 (TGFB1/2/3), Nodal and Activin A bind a dimer of type II serine/threonine kinase receptors, which in turn phosphorylate and activate two type I receptors, leading to the formation of a hetero-tetrameric receptor complex. Activation of the receptor complex leads to phosphorylation of SMAD2 and SMAD3, the receptor-SMADs (R-SMADs). Phosphorylated R-SMADs form heteromeric complexes with SMAD4 and translocate into the nucleus, where they bind target genes.

SMAD3 binds the DNA directly, while SMAD2 needs SMAD4 to do so[12–14], in combination with the histone acetyltransferase EP300, ultimately leading to activation of target genes. R-SMADs are known to interact with additional transcription factors that may vary between different cell types[15], resulting in activation of cell type-specific transcriptional programmes (see Supplementary Fig. 1a for a diagram of the TGF-beta pathway)[12,13]. In order to understand how TGF-beta signalling regulates the behaviour of hPSCs, it is critical to identify genes directly induced by R-SMADs.

Core pluripotency factors—POU5F1/OCT4, NANOG, SOX2— were initially identified in murine naïve pluripotent cells[16–18] and were then found to be functionally relevant in hPSCs[19]. A large set of additional murine pluripotency factors have been identified[20], the majority of which are not expressed in conventional human PSCs, potentially because of differences between species or because conventional hPSCs are in a more advanced developmental state called primed pluripotency. Although naïve hPSCs have been recently generated either directly from embryos or by reprogramming of somatic cells[21–24], they are not the focus of this study and, for clarity, we should stress that the acronyms hPSCs, hESCs and hiPSCs indicate only human conventional pluripotent cells in a primed state.

Here, we study conventional hPSCs with the aim of isolating human-specific pluripotency regulators that could reveal differences between PSCs of different species, or could play a critical role for induction of human pluripotency.

In this study, we characterise the transcriptional programme activated by TGF-beta/SMAD3 signalling in hPSCs. We identify several potential downstream mediators and test them using a gain-of-function approach. TGF-beta appears to maintain pluripotency via induction of four factors. Among them, we extensively characterise a transcriptional regulator, called ZNF398, which induces genes associated with pluripotency and epithelial character in collaboration with SMAD3 and the histone acetyltransferase EP300. Moreover, ZNF398 knockdown during somatic cell reprogramming causes a drastic reduction in iPSC colonies.

## Results

### Identification of TGF-beta transcriptional targets in hPSCs.
We expanded hPSCs under chemically defined conditions[4,5] and validated the known role of TGF-beta in maintenance of pluripotency using SB431542 (SB43), an inhibitor of TGFBR1 and ACVR1B/C, the type I receptors mediating TGFB1/2/3, Activin and Nodal signalling (Supplementary Fig. 1a). SB43 reduced phosphorylation of SMAD3 downstream of TGFB1 and reduced the levels of the pluripotency factors POU5F1/OCT4, PRDM14 and NANOG (Fig. 1a and Supplementary Fig. 1b) as previously reported[5–7].

Human pluripotent colonies are composed of a monolayer of cells expressing epithelial markers. SB43 treatment induces also a morphological change, with loss of cell–cell contact, reduction of epithelial markers and upregulation of mesenchymal markers (Fig. 1b and Supplementary Fig. 1c), as previously described[25].

We decided to study how TGF-beta controls gene programmes associated with pluripotency and the epithelial character with an unbiased functional approach based on the identification of direct transcriptional targets followed by functional validation[26,27]. We reasoned that TGF-beta transcriptional targets should be bound by SMAD2/3 and either downregulated upon signal inhibition or rapidly induced upon stimulation. SMAD2 and SMAD3 can also form heterodimers[14] and have redundant functions in pluripotent cells[28]. We focused on SMAD3, given that it is more abundant than SMAD2 in hPSCs and it binds directly to the DNA[12,13] (Supplementary Fig. 1d). We intersected SMAD3 chromatin immunoprecipitation followed by high-throughput sequencing (ChIP-seq) with gene expression data from cells treated with SB43 and identified 195 genes downregulated and bound by SMAD3 (Fig. 1c and d, top panel yellow dots on the left). Moreover, by intersecting SMAD3-bound genes with genes induced after 4 h of acute stimulation, we identified 61 additional putative targets and 20 that were also among the downregulated genes (Fig. 1c and d, bottom panel yellow dots on the right). Several known TGF-beta direct targets, such as LEFTY1/2, SKIL and SMAD7, were identified (Fig. 1d), supporting the validity of our approach.

We then refined our gene list by focusing on genes encoding for transcriptional regulators, such as transcription factors or chromatin modifiers, given that such classes of proteins have the capacity to direct transcriptional programmes. Finally, we included only genes robustly expressed (>3 RPKM in hPSCs) (Supplementary Fig. 1e), obtaining a list of 21 candidates (for all datasets, see Supplementary Data 1).

We performed qPCR to independently validate our putative TGF-beta targets. In particular, we tested the responsiveness to both TGFB1 and Activin A, two ligands commonly used for hPSCs expansion[4–7] (Supplementary Fig. 1a and f). We also tested whether targets were responsive to the TGF-beta signal when cells were expanded either on feeders or under feeder-free conditions, given that the TGF-beta signal is active and maintains pluripotency under both conditions (Supplementary Fig. 1g)[5,8,9]. After extensive validation, we identified eight genes (ID1, MYC, BCOR, KLF7, OTX2, ZNF398, NANOG and ETS2) as bona fide TGF-beta and Activin A transcriptional targets in hPSCs (Fig. 2a, b).

### Functional identification of pluripotency regulators.
If a gene is a critical downstream mediator of the TGF-beta signal in hPSCs, its forced expression should maintain pluripotency also when TGF-beta signalling is inhibited. To test this hypothesis, we stably expressed our candidates in hiPSCs and hESCs using piggyBac (PB) vectors.

First, we performed a clonal assay, which allows us to quantify the fraction of hPSCs able to self-renew, giving rise to pluripotent colonies. Cells transfected with an empty vector formed a reduced number of alkaline phosphatase (AP)-positive pluripotent colonies when treated with SB43 (Fig. 3a and Supplementary

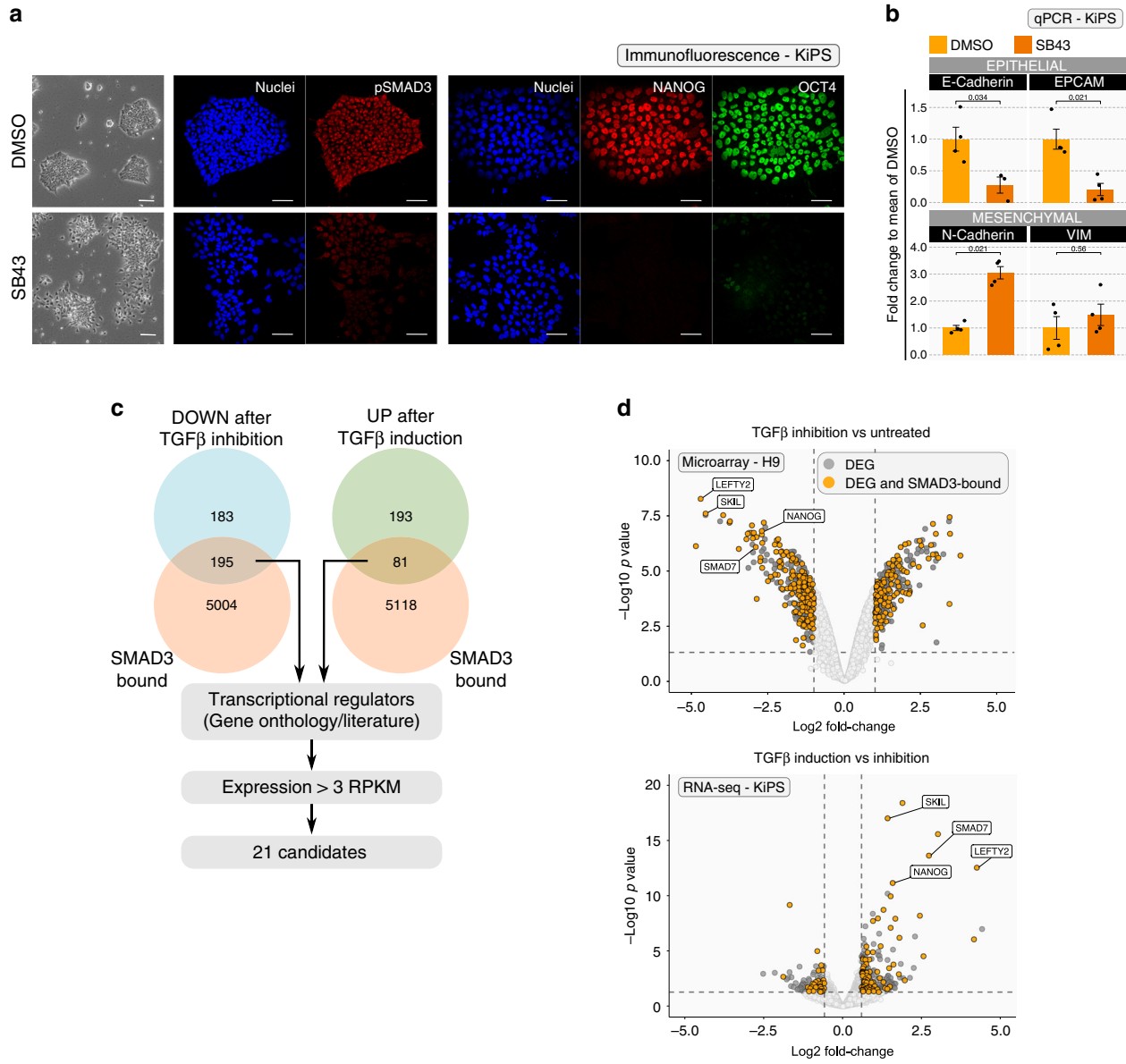

**Fig. 1 Identification of TGF-beta/SMAD3 transcriptional targets in hPSCs. a** Left: Morphology of KiPS treated with SB43 (10 μM) or the vehicle DMSO for 5 days. Right: Immunostaining for phosphorylated SMAD3 (pSMAD3) and the pluripotency markers NANOG and POU5F1/OCT4 shows a reduction of these markers after 5 days of SB43 treatment. See also Supplementary Fig. 1c for additional markers. Representative images of four independent experiments for NANOG and POU5F1/OCT4 and two independent experiments for pSMAD3 are shown. Scale bars 50 μm. **b** Gene-expression analysis by qPCR of KiPS treated with SB43 or DMSO for 5 days. Bars indicate the mean ± SEM (standard error of the mean) of four independent experiments shown as dots. Expression was normalised to the mean of DMSO samples. Unpaired two-tailed Mann–Whitney *U* test. Source data are provided as a Source Data file. **c** Approach used to identify potential SMAD3 direct targets. See also Supplementary Fig. 1e. **d** Top: Transcriptome analysis of hESCs treated with SB43 for 48 h (microarray data from ref. [10]). Dark grey dots indicate differentially expressed genes (DEGs) for −1 > Log2 fold-change > 1 and p-value < 0.05. Orange dots refer to DEGs bound by SMAD3 (data from ref. [15]), max distance between peak midpoint and TSS: ±50 kb. p-values were calculated with limma (v3.18.13)[64] and were adjusted for multiple testing with Benjamini–Hochberg correction. Bottom: Transcriptome analysis of hiPSCs treated for 4 h with mTeSR after 16 h of SB43 treatment (RNA-seq data, in this study). Dark grey dots indicate DEGs for −0.585 > Log2 fold-change > 0.585 (corresponding to an increase of 50%) and p-value < 0.05. Orange dots refer to DEGs bound by SMAD3. Known SMAD3 targets, such as NANOG, LEFTY2, SKIL and SMAD7 serve as positive controls[10,11,13]. p-values were calculated with edgeR package (v3.4.2)[60] and were adjusted for multiple testing with Benjamini–Hochberg correction.

Fig. 2a). Only expression of NANOG, KLF7, MYC and ZNF398 resulted in full rescue in formation of AP-positive colonies in the presence of SB43, while other factors had only a partial, or no effect. Second, under forced expression of either NANOG, KLF7, MYC or ZNF398 the cells maintained a flat epithelial-like morphology, generally associated with pluripotency (Fig. 3b), upon SB43 treatment, while other factors failed to do so. Third, NANOG, KLF7, MYC and ZNF398 were each able to maintain

expression of pluripotency markers (Fig. 3c) in presence of SB43, although they displayed specificity for different targets. For instance, ZNF398 and NANOG activated robustly PRDM14 expression. Quantitative immunostaining confirmed maintenance of OCT4 and NANOG at the protein levels (Fig. 4a). Comparable results were also obtained after prolonged culture with SB43 (Supplementary Fig. 2b–d). We confirmed our results in an additional hESC line (Supplementary Fig. 3a–c) and confirmed a

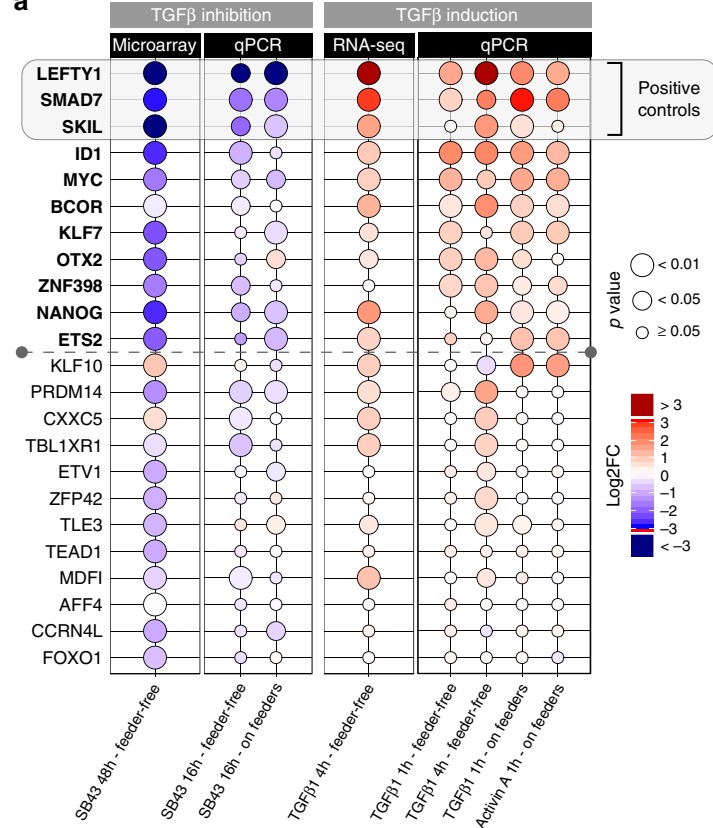

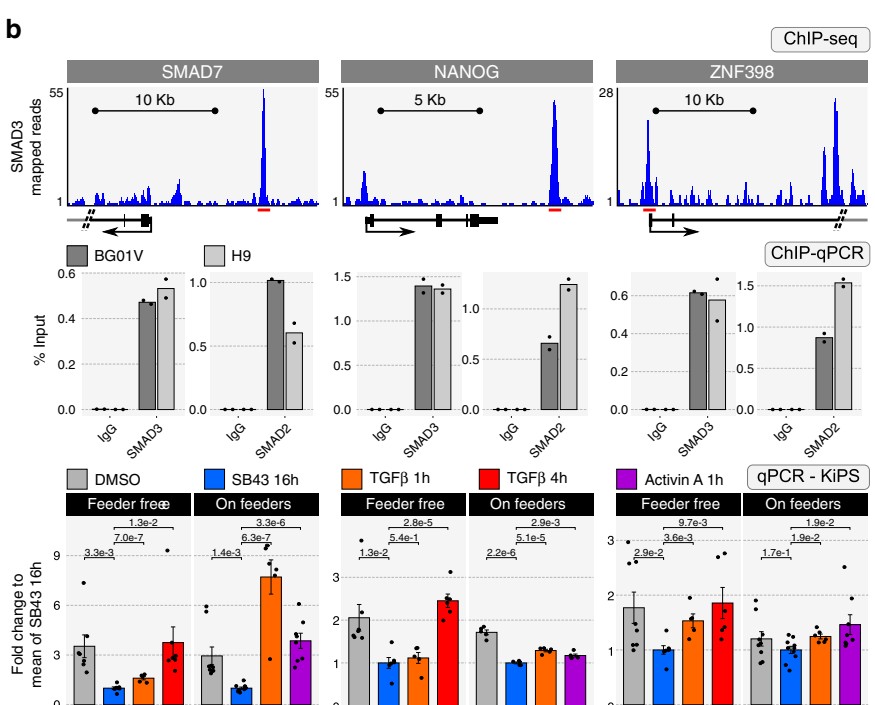

similar level of transgene expression in different cell lines for different constructs (Supplementary Fig. 3d).

Taken together these results indicate that forced expression of NANOG, KLF7, MYC and ZNF398 is individually sufficient to stably maintain an ESC-like state upon TGF-beta inhibition. Thus, we identified from the literature an extended suite of functional regulators of human pluripotency: FOXO1[29], PRDM14[30], BCOR[31],

LIN28A[3], LIN28B[32], DPPA2/4[33], SOX2[19], UTF1[34] and, in the present work, we identified the transcription factors KLF7 and ZNF398.

We noticed that MYC had a strong effect on AP-positive colony formation and morphology despite the partial effects on OCT4 and NANOG expression (Figs. 3a, b and 4a), suggesting that MYC might maintain pluripotency via other pluripotency factors. We

**Fig. 2 Validation of TGF-beta transcriptional targets. a** Balloon plot summarising validation experiments. Balloon size indicates the statistical significance, colour indicates the fold-change in expression relative to DMSO-treated hPSCs. Left: Microarray data as in top panel of Fig. 1d were independently validated by qPCR. Validation experiments were performed in two different culture conditions: feeder-free or on feeders (MEF). Expression was normalised to the mean of DMSO-treated cells. Right: RNA-seq data as in bottom panel of Fig. 1d were independently validated by qPCR. Expression was normalised to the mean of SB43-treated samples. The grey box highlights the known direct targets of SMAD3 (positive controls, including SKIL, one of the 21 candidates). Transcriptional SMAD3 targets independently confirmed are highlighted in bold. For qPCR validation, five independent experiments were performed for each condition. Unpaired two-tailed *t*-test, *p*-values were not adjusted. See also Supplementary Fig. 1f, g. Source data are provided as a Source Data file. **b** Example of SMAD3 binding and gene expression analysis of three validated targets. Top: Gene tracks represent binding of SMAD3 (data from ref. [15]) at the indicated gene loci. Red lines indicate the regions validated by ChIP-qPCR. Center: ChIP-qPCR on SMAD7, NANOG and ZNF398 loci was performed using anti-SMAD3 and anti-SMAD2 or a rabbit control IgG antibody in BG01V (dark grey) and H9 (light grey) cell lines. Enrichment is expressed as a percentage of the DNA inputs. Bars indicate the mean of two biological replicates shown as dots. Bottom: Gene expression analysis by qPCR. Bars indicate mean ± SEM of *n* = 7, 7, 5, 7, 9, 10, 8, 6 biological replicates over six independent experiments shown as dots for SMAD7, *n* = 7, 6, 5, 6, 5, 5, 5 biological replicates over five independent experiments shown as dots for NANOG, and *n* = 8, 7, 5, 6, 9, 10, 8, 6 biological replicates over six independent experiments shown as dots for ZNF398. Expression was normalised to the mean of SB43 16 h samples. Unpaired two-tailed *t*-test. Source data are provided as a Source Data file.

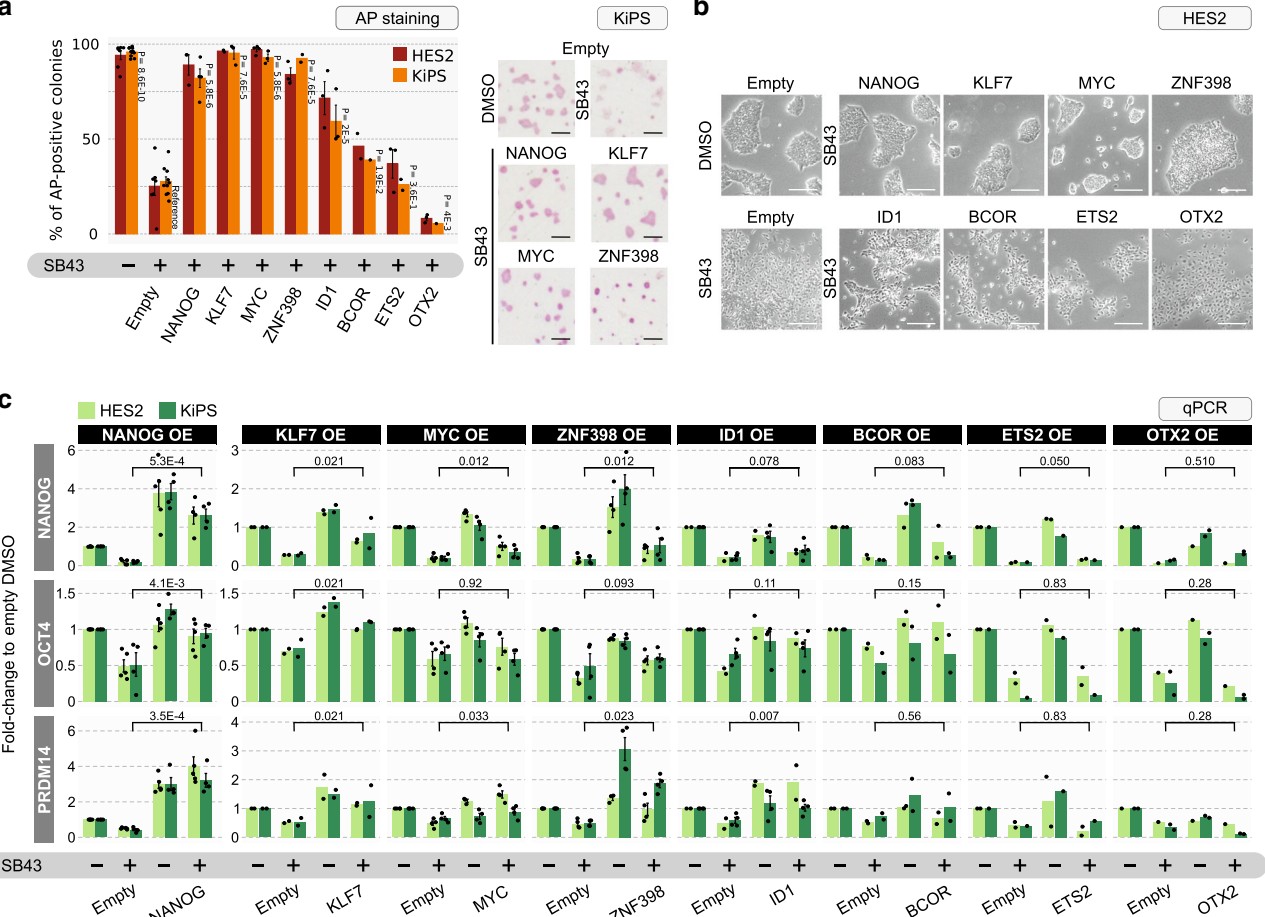

**Fig. 3 Functional identification of pluripotency regulators in hPSCs. a** Left: Clonal assay quantification of hESCs (HES2, red bars) and hiPSCs (KiPS, orange bars) stably expressing an empty vector control (Empty) or eight different SMAD3 targets identified in Fig. 2a. Two thousand cells were seeded at clonal density in the presence of DMSO or SB43 and stained for alkaline phosphatase (AP) after 5 days. Bars show the mean ± SEM percentage of AP-positive colonies. Dots represent independent experiments (*n* = 7, 10 for Empty DMSO; *n* = 7, 10 for Empty SB43; *n* = 3, 4 for NANOG SB43; *n* = 2, 2 for KLF7 SB43; *n* = 4, 3 for MYC SB43; *n* = 3, 2 for ZNF398 SB43; *n* = 3, 3 for ID1 SB43; *n* = 2, 1 for BCOR SB43; *n* = 3, 2 for ETS2 SB43; *n* = 3, 1 for OTX2 SB43 in HES2 and KiPS, respectively). Unpaired two-tailed Mann–Whitney *U* test relative to Empty SB43 samples. Right: Representative images of clonal assay performed in KiPS. See also Supplementary Fig. 3a for results obtained in H9 hESCs. Scale bars 500 μm. Source data are provided as a Source Data file. **b** Morphology of HES2 colonies stably expressing an empty vector (Empty) in presence of DMSO or SB43 and HES2 stably expressing the eight SMAD3 targets in presence of SB43. Representative images of three independent experiments are shown. See also Supplementary Fig. 3b for results obtained in H9. Scale bars 200 μm. **c** Gene expression analysis by qPCR of HES2 (light green bars) and KiPS (dark green bars) stably expressing an Empty vector or the eight SMAD3 targets and treated with or without SB43 for 5 days. Bars indicate mean ± SEM of independent experiments, shown as dots (*n* = 5, 5 for NANOG overexpression; *n* = 2, 2 for KLF7 overexpression; *n* = 4, 4 for MYC overexpression; *n* = 4, 4 for ZNF398 overexpression, *n* = 2, 4 for ID1 overexpression; *n* = 2, 2 for BCOR overexpression; *n* = 2, 1 for ETS2 overexpression; *n* = 1, 2 for OTX2 overexpression in HES2 and KiPS, respectively). Expression was normalised to the Empty DMSO samples. Unpaired two-tailed Mann–Whitney *U* test. Source data are provided as a Source Data file.

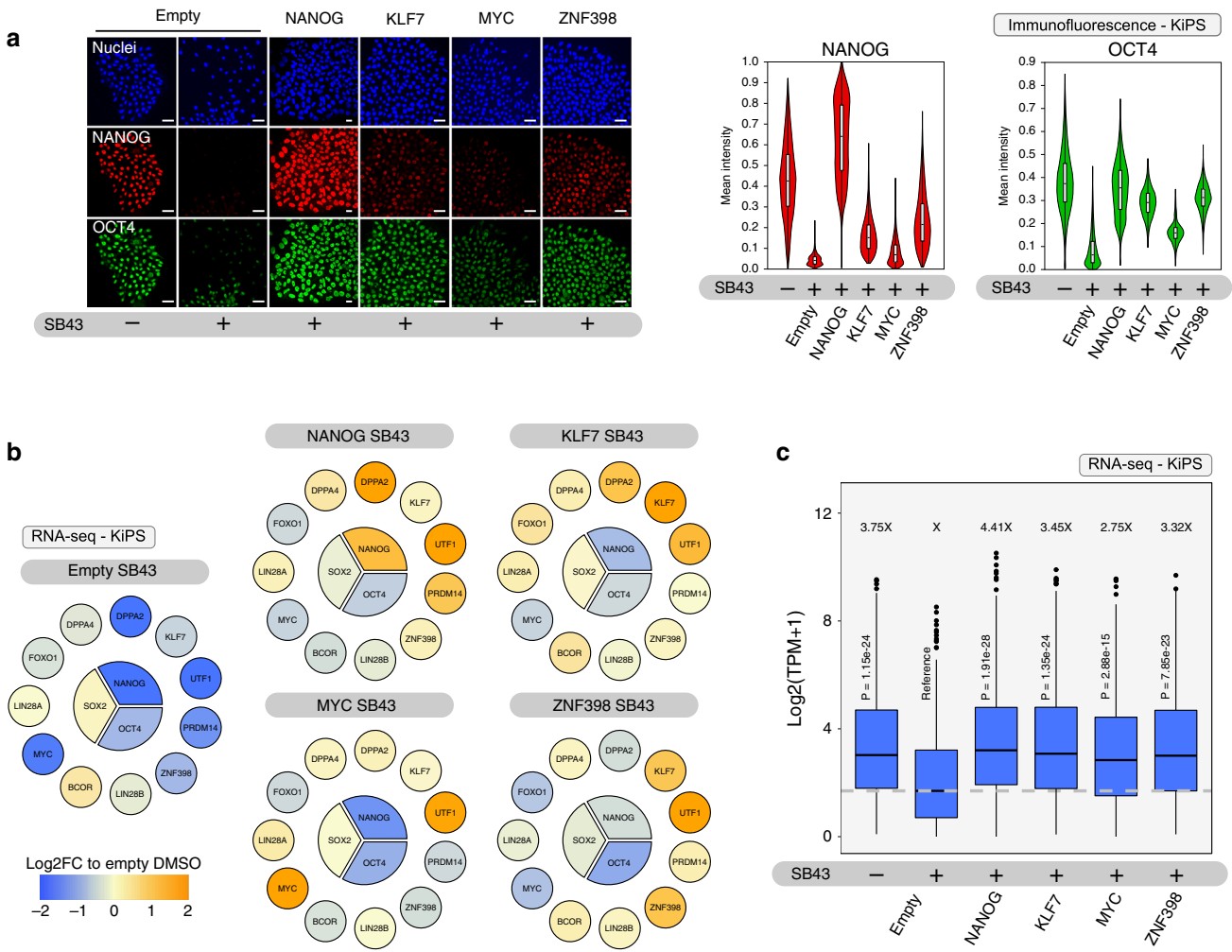

**Fig. 4 A quartet of transcriptional regulators maintain pluripotency. a** Left: immunostaining for the pluripotency markers NANOG and POU5F1/OCT4 of KiPS stably expressing an empty vector control (Empty) in presence of DMSO or SB43 and KiPS stably expressing NANOG, KLF7, MYC or ZNF398 in presence of SB43 for 5 days. Representative images of three independent experiments are shown. Right: Violin plots showing fluorescence intensity quantification of NANOG and OCT4. For each condition, at least 1200 nuclei from five randomly selected fields were analysed. Box plot indicates 25th, 50th and 75th percentile; whiskers indicate minimum and maximum. Scale bars 20 μm. See also Supplementary Fig. 3c for results obtained in H9. Source data are provided as a Source Data file. **b** Diagrams showing an extended set of pluripotency regulators. Gene expression analysis by RNA-seq of KiPS stably expressing an empty vector, NANOG, KLF7, MYC or ZNF398 and treated with SB43 for 5 days. Colours indicate the fold-change relative to Empty DMSO sample, thus yellow indicates the endogenous expression of a given gene in undifferentiated hPSCs. **c** Box plot showing absolute expression levels (normalised counts, TPM) of 538 genes DOWN-regulated by SB43 treatment (5 days) in KiPS stably expressing an empty vector (see Fig. 5a, blue dots). Shown data refers to KiPS transfected with the empty vector in the presence of DMSO or SB43 (n = 4, 4 independent experiments, respectively) and for KiPS stably expressing NANOG, KLF7, MYC or ZNF398 in presence of SB43 for 5 days (n = 2, 2, 2, 2 independent experiments, respectively). Average fold-change relative to Empty SB43 sample (X) is reported for each condition. Box plots indicate 25th, 50th and 75th percentile; whiskers indicate minimum and maximum. Unpaired two-tailed t-test.

performed transcriptome analysis and observed that MYC robustly activated UTF1, KLF7, LIN28B and DPPA2, despite the mild effect on NANOG and OCT4 (Fig. 4b). Similarly, NANOG, KLF7 and ZNF398 activated completely distinct sets of pluripotency regulators. We further extended our analysis to all genes highly expressed in hPSCs that were significantly reduced upon SB43 treatment for 5 days. We identified 538 genes downregulated, as a proxy for genes generally associated with human pluripotency (Fig. 5a, mean fold-reduction relative to Empty-DMSO and Empty-SB43 = 3.75×), among them were also PRDM14 and NANOG. All of our four factors under study were able to rescue such global transcriptional effect (Fig. 4c, see data in Supplementary Data 2).

Murine epiblast stem cells (EpiSCs) are primed pluripotent cells derived from the post-implantation epiblast[35,36]. EpiSCs share several molecular features with primed hPSCs[37], including

the requirement of TGF-beta for self-renewal[10]. Therefore, we asked whether forced expression of the four factors would maintain pluripotency also in EpiSCs. We generated both GOF18[27] and OEC2[38] EpiSCs stably expressing the four transcription factors (Supplementary Fig. 4a). TGF-beta inhibition led to a reduction of Nanog, Oct4, Otx2 and Fgf5 (Supplementary Fig. 4b) and none of the four factors were able to maintain the expression of the markers analysed, with the exception of Otx2 maintained only by KLF7. We conclude that the ability of NANOG, KLF7, MYC and ZNF398 to maintain pluripotency is not conserved in murine EpiSCs.

In sum, our results indicate that in hPSCs, TGF-beta maintains pluripotency mainly via a quartet of transcriptional regulators, each one preferentially activating a specific subset of pluripotency factors. Among these, NANOG and MYC have

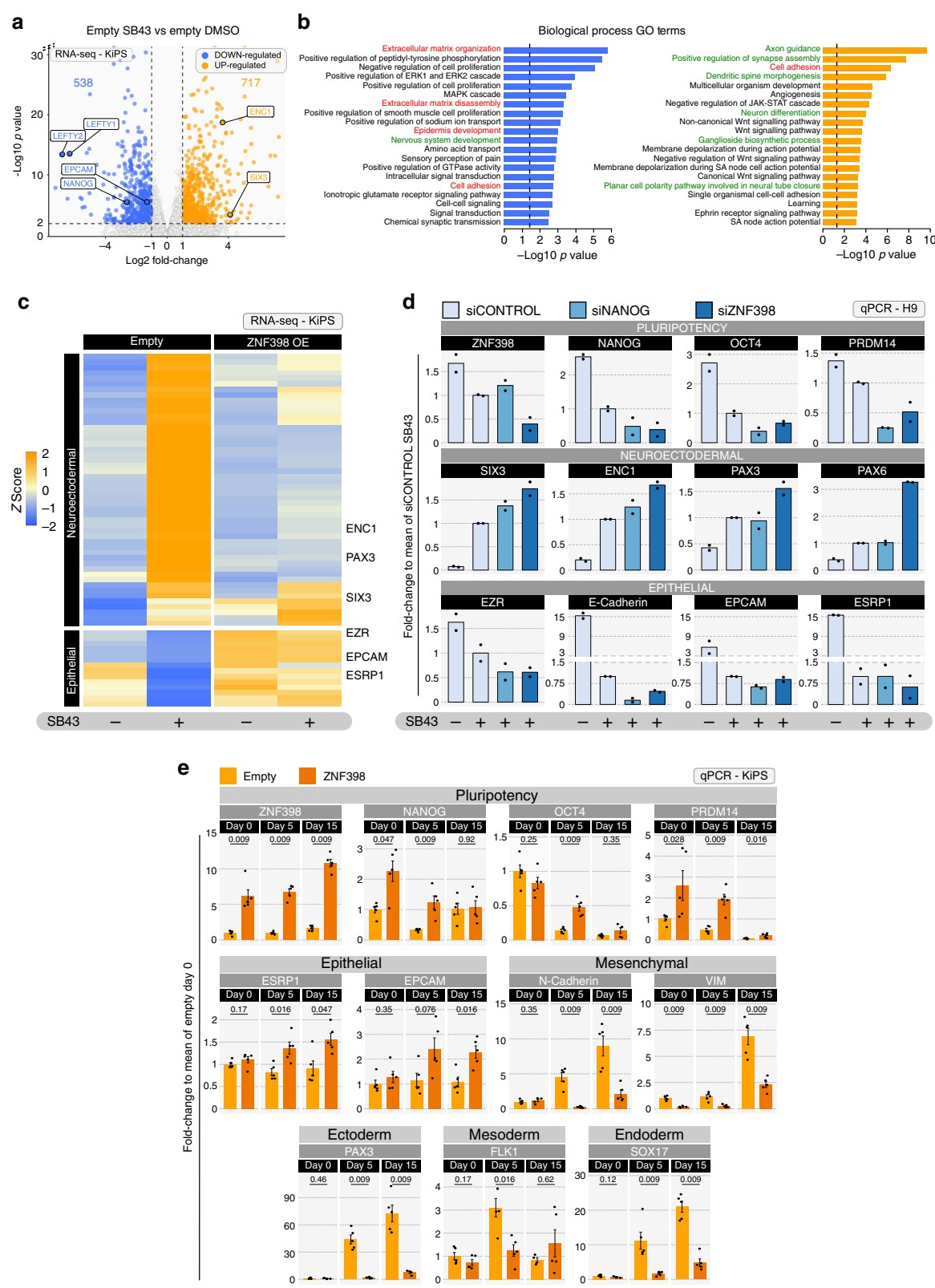

been extensively investigated as pluripotency regulators[10,11,19,39]; KLF7 is a Kruppel-like factor and other members of the same family, such as KLF2/4/5, are known regulators of pluripotency[40]. Conversely, ZNF398 has never been implicated in regulation of pluripotency, prompting us to choose it for further molecular characterisation.

**ZNF398 represses differentiation and mesenchymal genes.** When TGF-beta is blocked hPSCs lose pluripotency and undergo a morphological change. After focusing on the pluripotency regulators (Figs. 3 and 4), we decided to study the global effect of TGF-beta on hPSC function. Thus, we performed an unbiased transcriptional analysis and observed that upon SB43 treatment

**Fig. 5 ZNF398 represses differentiation and mesenchymal genes. a** Transcriptome analysis of KiPS stably expressing an empty vector and treated with SB43 for 5 days. DOWN-regulated (Log2 fold-change < −1 and p-value < 0.01) and UP-regulated (Log2 fold-change > 1 and p-value < 0.01) genes are indicated in blue and orange, respectively. Known TGF-beta targets (LEFTY1, LEFTY2) serve as controls. Not adjusted p-values were calculated with Wald Test. **b** GO term analysis for biological processes of DOWN-regulated genes (left, blue bars) and UP-regulated genes (right, orange bars) revealed a statistically significant enrichment (p-value < 0.05) for genes involved in cell adhesion, epithelial to mesenchymal transition, organisation of extracellular matrix (highlighted in red) and neural development (highlighted in green). p-values were calculated by Fisher Exact test using DAVID database[63]. **c** Heatmap for markers of neuroectodermal and epithelial character. RNA-seq data derived from KiPS stably expressing an empty vector (Empty) or ZNF398 and treated with DMSO or SB43 for 5 days. Z-scores of row-scaled expression values (TPM) are shown. Orange and blue indicate high and low expression, respectively. Markers of epithelial (EPCAM, ESRP1, EZR) and neuroectodermal identity (SIX3, ENC1, PAX3) are highlighted. **d** Gene expression analysis by qPCR of H9 transfected with the indicated siRNAs (a non-targeting siRNA (siCONTROL) or a pool of two validated siRNAs (siNANOG and siZNF398)) and treated with or without SB43 for 5 days. See also "Methods" section. Bars indicate the mean of two independent experiments shown as dots. Expression was normalised to the mean of siCONTROL treated with SB43 samples. See also Supplementary Fig. 5 for siRNA validation and for additional markers. Source data are provided as a Source Data file. **e** Gene expression analysis by qPCR of EBs differentiation of KiPS stably expressing an empty vector (Empty, light orange) or ZNF398 (dark orange) analysed at three different time points (Day 0, Day 5 and Day 15) of differentiation. Bars indicate the mean ± SEM of five independent experiments, shown as dots. Expression was normalised to the mean of Empty Day 0 sample. Unpaired two-tailed Mann–Whitney U test. Source data are provided as a Source Data file.

538 genes were downregulated and 717 were upregulated (Fig. 5a). Gene Ontology (GO) enrichment analysis identified several categories associated with cell adhesion, epithelial to mesenchymal transition and organisation of the extracellular matrix, in agreement with the observed morphological change (Fig. 5b). Among them we identified a subset of genes specifically associated with epithelial character, that were downregulated by SB43 (Fig. 5c, epithelial). Moreover, we observed several gene categories associated with formation and function of neural cells (Fig. 5b), corresponding to a set of genes upregulated by SB43 (Fig. 5c, neuroectodermal). Upregulation of neuroectodermal genes was expected from studies performed in different model systems showing that TGF-beta, Activin A and Nodal block neuroectoderm formation[41,42]. Indeed, inhibition of TGF-beta is commonly used for neuroectodermal differentiation protocols[43].

Next, we asked whether the forced expression of ZNF398 was able to counteract such transcriptional changes and observed a reduction in neuroectodermal genes and boosted expression of epithelial genes (Fig. 5c). We conclude that ZNF398 is activated by TGF-beta to maintain the correct expression of neuroectodermal and epithelial genes in hPSCs.

Next, we asked whether ZNF398 would be required to control TGF-beta-dependent transcriptional programmes. We performed siRNA-mediated knockdown of ZNF398 and observed no effect on self-renewal (Supplementary Fig. 5a, b), as expected from the presence of four factors that are individually able to maintain pluripotency downstream of TGF-beta. However, ZNF398 knockdown during the early phases (5 days) of differentiation resulted in further reduction of pluripotency and epithelial markers, and enhanced induction of neuroectodermal genes (Fig. 5d, see also Supplementary Fig. 5c) to an extent comparable or greater than NANOG knockdown.

To further investigate the capacity of ZNF398 to regulate pluripotency and the epithelial character of hPSCs in an independent assay, we performed embryoid bodies (EBs) differentiation. Forced expression of ZNF398 was able to activate expression of pluripotency and epithelial markers, while repressing mesenchymal and germ layer markers relative to control cells (Fig. 5e). Collectively, these results indicate that ZNF398 promotes the expression of pluripotency and epithelial markers and represses genes associated with differentiation of hPSCs.

## ZNF398 activates transcription in concert with SMAD3.

We next sought to understand the molecular mechanism by which ZNF398 promotes pluripotency and epithelial character. ZNF398 contains several zinc-finger domains and it has been shown to recognise specific DNA sequences in COS-1 cells[44]. Therefore, we

performed ChIP-seq for ZNF398 in two different hESCs lines and identified genomic regions bound by it containing a DNA motif similar to other ZNF factors (Supplementary Fig. 6a). Cooperative binding among transcription factors has been reported in several stem cell systems, thus we asked how similar the genome-wide-binding profile of ZNF398 is to those of other transcriptional regulators (data from CODEX[45]). Surprisingly, we found that ZNF398 clustered more closely with SMAD3 and the histone acetyl-transferase EP300, compared to the core pluripotency factors OCT4/NANOG or the Polycomb components (Fig. 6a, top panel). A similar analysis conducted on histone modifications-associated ZNF398 with regions decorated by acetylation of histone 3 on lysine 9 or 27 (Fig. 6a, bottom panel). Clustering results are confirmed by strong colocalisation of ZNF398, SMAD3, EP300 and H3K27ac (Fig. 6b). Histone acetylation is associated with both active promoters and enhancers, so we looked at the distribution of mono-methylation and tri-methylation of histone 3 on lysine 4, associated with active enhancers and promoters, respectively. We looked at ZNF398 peaks and found that 3595 ZNF398 peaks out of 5771 appeared as active enhancers (high levels of H3K4me1 and low H3K4me3), while the remaining 2176 peaks as active promoters (high H3K4me3). We conclude that ZNF398 preferentially colocalises with SMAD3 and EP300 at active enhancers and promoters in hPSCs.

The frequent colocalisation may be due to binding to neighbouring DNA regions or to physical interaction. A co-immunoprecipitation (Co-IP) assay indicates that SMAD3 and ZNF398 form a complex in hPSCs (Fig. 6c).

We observed that ZNF398 bound and activated the pluripotency factor LIN28B and the epithelial master regulator epithelial splicing regulatory protein 1 (ESRP1[46]) (Fig. 7a), matching the pro-pluripotency and pro-epithelial activity of ZNF398. Interestingly, we also observed that LEFTY1, a known TGF-beta direct target, was also co-bound by ZNF398 (Fig. 7a). We therefore hypothesised that ZNF398 might potentiate the transcription of TGF-beta targets by binding SMAD3 targets. We functionally tested this hypothesis by comparing hPSCs expressing ZNF398 against control hPSCs. ZNF398 boosted the basal expression of LEFTY1 by >10 fold (i.e. in the presence of TGF-beta), and was even able to maintain residual LEFTY1 expression in the absence of TGF-beta signalling (Fig. 7b). We extended our analysis to all SMAD3 direct target genes (Fig. 1c) and observed that 23 out of 81 were also co-bound by ZNF398 (Fig. 7c, enrichment of 3.67 fold over those expected by chance, p-value = 3.49e−12, Chi-squared test). Importantly, the entire set of SMAD3-ZNF398 co-bound genes were significantly upregulated in cells expressing ZNF398 (Fig. 7d), further indicating a functional role of ZNF398 as activator of SMAD3 co-bound targets. This activity is specific

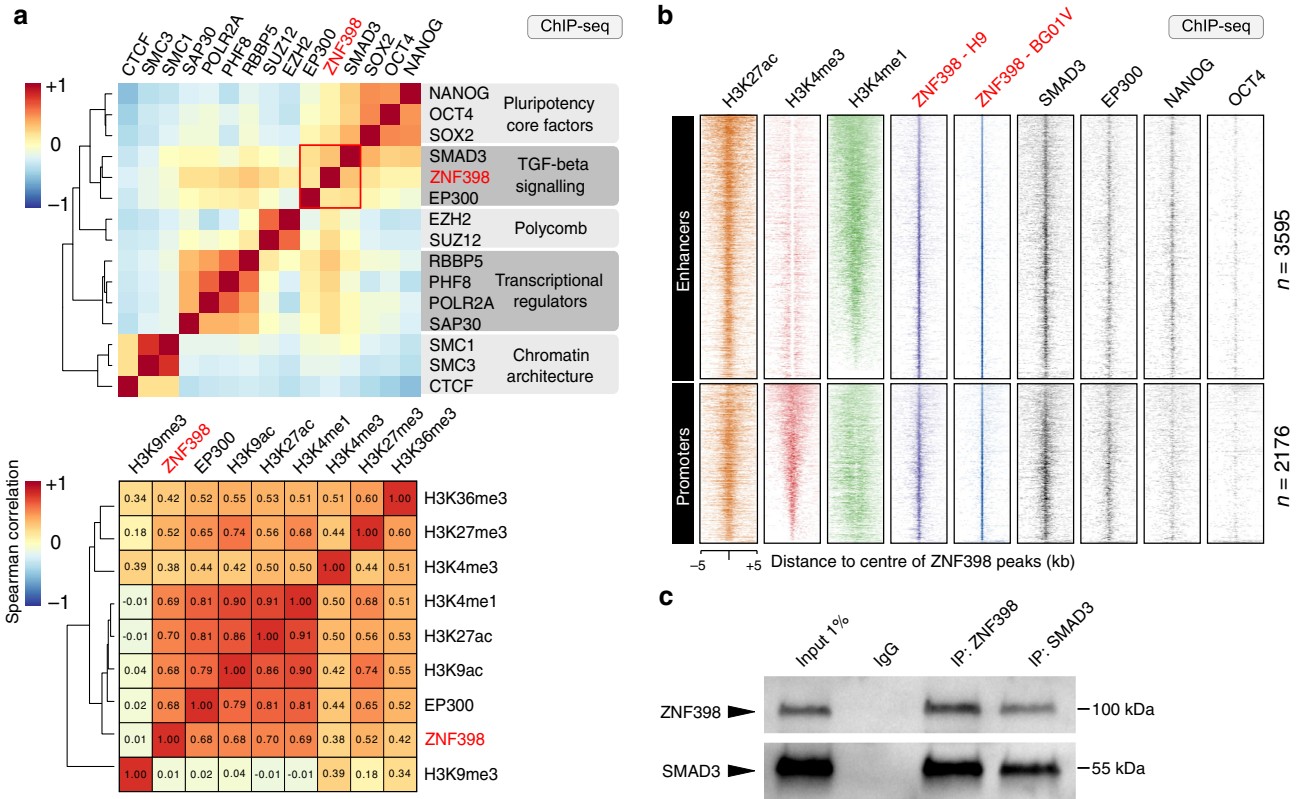

**Fig. 6 ZNF398 activates transcription in concert with SMAD3 and EP300. a** Top: Hierarchical clustering of 15 genome-wide-binding profiles (available genome-wide binding profiles from CODEX[45]). Normalised Pointwise Mutual Information (NPMI) between each pair of samples were used to display all pairwise binding overlaps in a clustered heatmap[45]. Colours in the heatmap show the level of overlap for each pair of samples (red, all binding sites overlapped; yellow, overlap expected by chance; blue, mutually exclusive binding). Bottom: Hierarchical clustering of pairwise Spearman correlation of ZNF398 and ChIP-seq datasets indicated. Colours indicate the level of correlation (red indicates perfect correlation, blue indicates perfect anticorrelation). ZNF398 clusters together with active histone marks. See also Supplementary Fig. 6a for DNA motifs associated with ZNF398 peaks. **b** Binding plots show the location of ZNF398 (obtained in two hPSC lines, H9 and BG0V1) and H3K27ac, H3K4me3, H3K4me1, SMAD3, EP300, NANOG and OCT4. 5771 sites are displayed within a 10 kb window centred around ZNF398 peaks. Note the presence of ZNF398 both at active enhancers (H3K4me1 positive) and active promoters (H3K4me3 positive). **c** SMAD3 interacts with ZNF398. Co-IP with antibodies against SMAD3, Avi-Tag-ZNF398 and IgG were performed on nuclear lysates of H9 expressing ZNF398, treated with 25 ng/ml Activin A for 1 h to promote nuclear accumulation of SMAD3. Precipitated complexes were probed for endogenous SMAD3 and Avi-Tag-ZNF398. Representative of two independent experiments. Uncropped gels are provided as a Source Data file.

for ZNF398, given that NANOG expression had no discernible effect on SMAD3-ZNF398 targets. Among the genes upregulated in hPSCs expressing ZNF398, we observed strong induction of several established direct targets of TGF-beta signal, such as *LEFTY1/2, CER1, TGFB1* and *NODAL* (Fig. 7e). We also analysed the dynamics of R-SMADs nuclear entry upon TGF-beta stimulation. Sixty minutes of treatment were sufficient to induce phosphorylation of SMAD3 and translocation from the cytoplasm to the nucleus (Supplementary Fig. 6b–d). Ectopic ZNF398 expression led to accelerated and enhanced nuclear translocation.

In sum, we conclude that ZNF398 colocalises with SMAD3 at active enhancers and promoters, activating the transcription of TGF-beta targets in hPSCs.

ZNF398 could be either a hPSC-specific or a general activator of the TGF-beta signal. We identified only two human cell lines expressing ZNF398 comparably to hPSCs (Supplementary Fig. 7a) and performed ZNF398 downregulation or over-expression, observing no differences in the induction of TGF-beta direct targets (Supplementary Fig. 7b, c). In two EpiSCs lines, stable expression of *Zfp398*—the *ZNF398* mouse orthologue—we also observed no effect on the levels of TGF-beta targets (Supplementary Fig. 7d), in stark contrast with what we observed in hPSCs expressing ZNF398. We conclude that, among all the cell types we tested, ZNF398 activates the TGF-beta signal only in hPSCs.

**ZNF398 is required for somatic cell reprogramming.** So far, our results indicate that ZNF398 promotes the pluripotency and the epithelial character programmes in hPSCs. We decided to test the function of ZNF398 in an orthogonal system, the induction of pluripotency from somatic cells. Reprogramming from somatic cells, such as fibroblasts, requires an early mesenchymal to epithelial transition (MET) followed by the activation of endogenous pluripotency factors[23,47,48]. We noticed that ZNF398 is expressed in human fibroblasts (Supplementary Fig. 8a), raising the possibility that ZNF398 promotes acquisition of epithelial character and pluripotency from early stages of reprogramming. We reprogrammed human fibroblasts by delivery of mRNAs encoding for either OSKMNL[23,47,48] (OCT4, SOX2, KLF4, MYC, NANOG, LIN28A) or OSKM, in combination with siRNAs, allowing to test the requirement of endogenous ZNF398 for reprogramming (Fig. 8a). By day 6 of reprogramming, fibroblasts transfected with control siRNAs formed clusters of epithelial cells, indicative of MET. This effect was clearly reduced upon ZNF398 knockdown (Fig. 8b and Supplementary Fig. 8b). Around day 10 small colonies emerged and were stabilised over the following 6 days. Upon Control siRNA and OSKMNL transfection we obtained 0.9% of reprogramming efficiency, which was reduced to 0.15% by ZNF398 knockdown. In the case of Control siRNA and OSKM the efficiency was 0.5% and ZNF398 knockdown almost

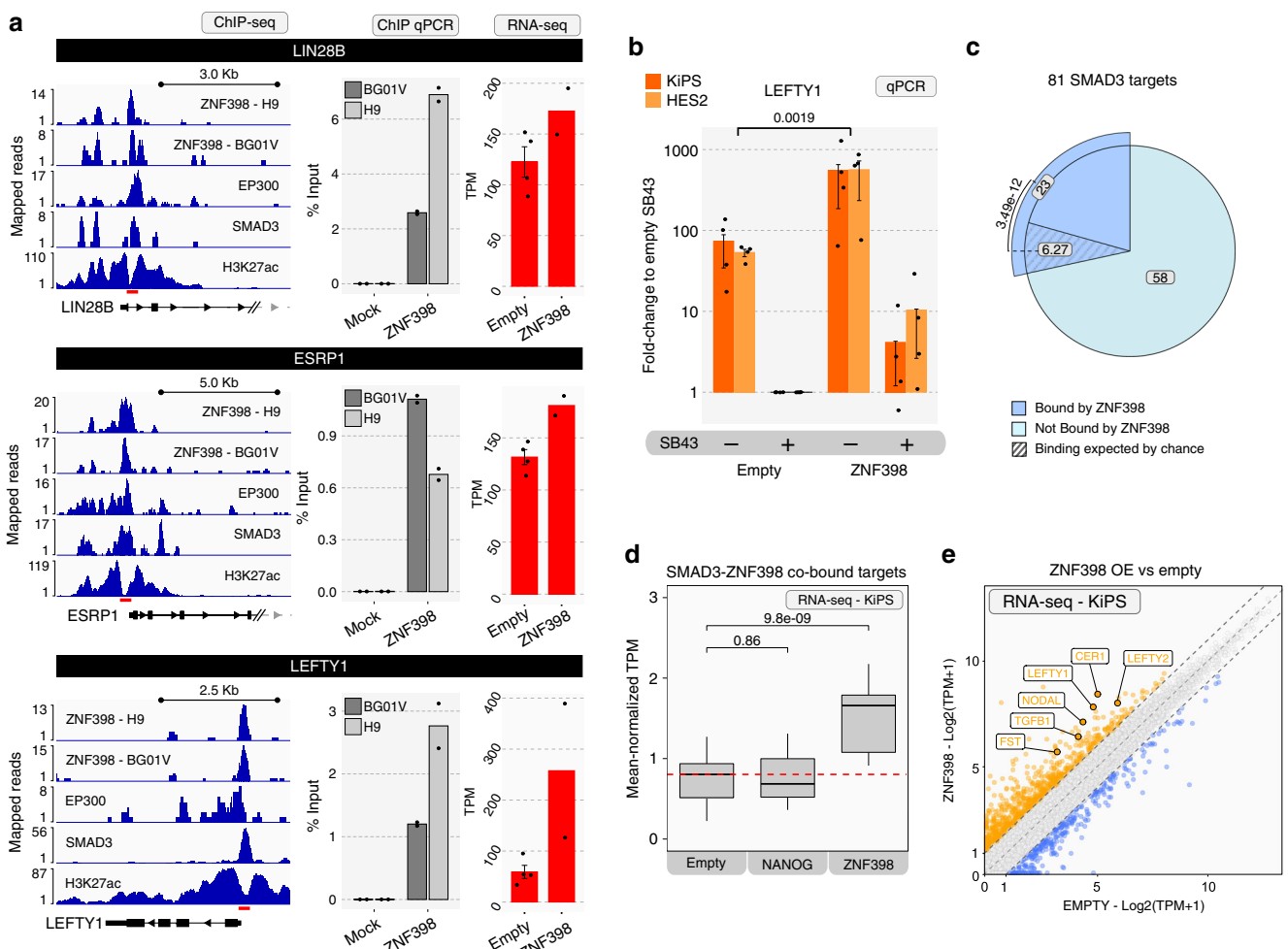

**Fig. 7 ZNF398 boosts TGF-beta signal. a** Left: Gene tracks of ZNF398 in two different hPSC lines (H9 and BG01V), EP300, SMAD3, and H3K27ac. Middle: Barcharts of ChIP-qPCR for ZNF398 performed in BG01V (dark grey) and H9 (light grey) cell lines on LIN28B, ESRP1 and LEFTY1 loci. Enrichment is expressed as a percentage of the DNA inputs. Bars indicate the mean of two biological replicates shown as dots. Right: RNA-seq of hPSCs (KiPS) stably expressing an empty vector (Empty) or ZNF398. Bars indicate the mean ± SEM of independent experiments shown as dots ($n = 4$ and $n = 2$ in Empty and ZNF398 overexpressing cells, respectively). Absolute expression is reported as TPM. Source data are provided as a Source Data file. **b** LEFTY1 levels measured by qPCR of HES2 (light orange bars) and KiPS (dark orange bars) stably expressing an empty vector or ZNF398, untreated or treated with SB43 for 5 days. Bars indicate the mean ± SEM of independent experiments shown as dots ($n = 4$). Expression was normalised to the Empty SB43 samples and shown on a logarithmic scale. Unpaired two-tailed Mann–Whitney $U$ test. Source data are provided as a Source Data file. **c** Pie-chart representing the 81 SMAD3 direct targets UP-regulated by TGF-beta induction identified in Fig. 1c. In dark blue are shown the 23 SMAD3 targets that are also bound by ZNF398. Such co-binding is significantly higher ($p$-value = 3.49e−12, Chi-squared test) than the one expected by chance, shown as the slice filled with diagonal lines. **d** Mean-normalised expression levels of the 23 genes bound by ZNF398 and SMAD3. Shown data derived from RNA-seq analysis of KiPS stably expressing an empty vector, NANOG (serving as a control) or ZNF398. For each gene, data was normalised to the mean-expression across the three samples. Box plot indicates 25th, 50th and 75th percentile; whiskers indicate minimum and maximum. Unpaired two-tailed $t$-test. **e** Scatter plot showing RNA-seq data from KiPS stably expressing an empty vector (Empty) or ZNF398. DOWN-regulated (Log2FC < −1) and UP-regulated (Log2FC > 1) genes are indicated in blue and orange respectively.

completely ablated formation of NANOG and OCT4-expressing colonies (Fig. 8c, d).

Transcriptional analysis indicates a failure to activate a large panel of pluripotency and epithelial markers upon ZNF398 knockdown (Fig. 8e and Supplementary Fig. 8c). Immunostaining confirmed membrane localisation of E-cadherin only in NANOG-positive reprogrammed colonies (Fig. 8f), accompanied with loss of actin stress fibres, clearly visible in fibroblasts that failed to reprogramme.

We also asked whether ZNF398 might be required for the proliferation of fibroblasts, rather than for acquisition of pluripotency and epithelial character. However, we could not observe a reduction in cell number after 6 days of siZNF398 transfection and

levels of proliferation regulators were also unchanged (Supplementary Fig. 8d, e).

Thus, ZNF398 is required for efficient induction of epithelial character and pluripotency from fibroblasts.

## Discussion

TGF-beta signalling is critical for hPSC self-renewal[5–7]. The transcription factor NANOG was first identified in murine ESCs for its capacity to maintain pluripotency in the absence of exogenous signals[16]. Such activity was also found conserved in hPSCs and it was shown that TGF-beta directly induces NANOG expression in hPSCs[10,11]. However, an unbiased and systematic analysis of

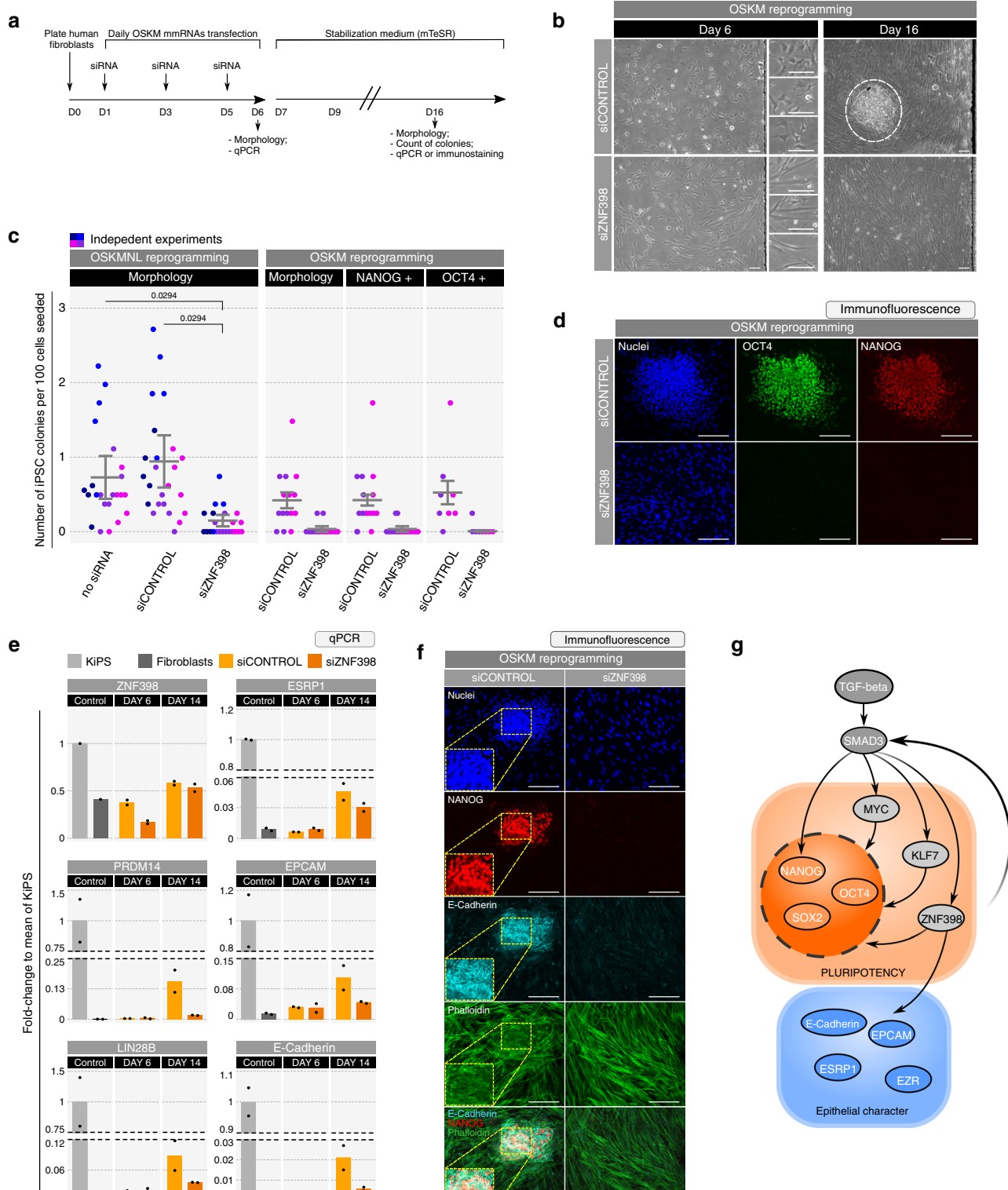

TGF-beta functional mediators in hPSCs was still missing. For this reason, we performed a transcriptome-level analysis of TGF-beta targets followed by a gain-of-function screening to identify uncharacterised pluripotency regulators.

Loss-of-function screenings have been performed in hPSCs, whereby genes were inactivated by RNA interference or using the CRISPR system[30,31,49]. Such studies identified some critical pluripotency regulators, such as PRDM14 or BCOR. However, loss-of-function approaches might fail to identify critical regulators because of functional redundancy with other factors. For example, a CRISPR screening in murine ESCs failed to identify the majority of known pluripotency factors[50], likely because the pluripotency network is highly redundant and robust to inactivation of single factors[20,40]. For this reason, we chose a gain-of-function screening approach, whereby individual putative pluripotency regulators are exogenously expressed in hPSCs and their capacity to maintain pluripotency is tested. Such an approach allowed the identification of several critical murine pluripotency regulators[20].

**Fig. 8 ZNF398 is required for somatic cells reprogramming. a** Experimental strategy for reprogramming by delivery of OSKM (OCT4, SOX2, KLF4, MYC) mRNAs in combination with siRNAs in order to test the requirement of ZNF398 for reprogramming. Scale bars 50 μm. See also "Methods" section for details. **b** Cell morphology during OSKM reprogramming. Representative images of two independent experiments are shown. See also Supplementary Fig. 8b. **c** Number of iPSC colonies obtained from 100 cells seeded at day 14 in OSKMNL reprogramming and at day 16 in OSKM reprogramming under the indicated conditions. Bars indicate the mean ± SEM of independent experiments. p-values: unpaired two-tailed Mann–Whitney U test. OSKMNL reprogramming: dots indicate biological replicates ($n = 23$, 24 and 23 replicates for no siRNA, siCONTROL and siZNF398, respectively) from four independent experiments shown in different shades of colours. OSKM reprogramming: dots indicate biological replicates ($n = 15$ for replicates scored based on morphology or NANOG signal, $n = 8$ for replicates scored based on OCT4 signal) from two independent experiments shown in different shades of colours. Source data are provided as a Source Data file. **d** Immunostaining for OCT4 and NANOG of fibroblasts transfected with OSKM mRNAs and siCONTROL or siZNF398 at day 16. Representative images of two independent experiments are shown. Scale bars 150 μm. **e** Gene expression analysis by qPCR of hiPSCs (KiPS) (light grey) and fibroblasts (dark grey) serving as controls, and fibroblasts transfected with OSKMNL mRNAs and siCONTROL (light orange) or siZNF398 (dark orange) at day 6 and day 14. Bars indicate the mean of two independent experiments shown as dots. Expression was normalised to the mean of KiPS samples. See also Supplementary Fig. 8a, c. Source data are provided as a Source Data file. **f** Immunostaining for NANOG and E-Cadherin followed by Phalloidin staining of fibroblasts transfected with OSKM mRNAs and siCONTROL or siZNF398 at day 16. Representative images of two independent experiments are shown. Scale bars 150 μm. **g** Diagram representing the transcription factors induced by TGF-beta, among which ZNF398 is crucial for the maintenance of pluripotency and the epithelial character of hPSCs.

We identified a quartet of transcription factors, NANOG, MYC, KLF7 and ZNF398, which individually promote hPSC self-renewal. Interestingly, each of these four factors activates a specific subset of human pluripotency regulators[3,19,29–34], indicating that the human pluripotency network is flexible and can be maintained under different configurations. Among them, ZNF398 controls both pluripotency and epithelial genes downstream of TGF-beta (Fig. 8g).

Our analyses identified an extended set of functional human pluripotency regulators beyond the core factors OCT4, SOX2 and NANOG (Fig. 4b). It will be interesting to apply computational modelling[20] to reconstruct the network of interactions among such factors in order to study how such a network reconfigures itself after perturbations or during reprogramming.

Interestingly, only a fraction of human pluripotency regulators are robustly expressed in murine ESCs (data from ref. [21]). This observation is in part attributable to differences in the developmental stage, as conventional hPSCs are in a pluripotent stage primed for differentiation, whereas murine ESCs are in a more primitive, naïve state of pluripotency[20,37].

However, naïve hPSCs have been recently obtained[21–24] and we observed that ZNF398 and KLF7 are robustly expressed in hPSCs regardless of their pluripotency state. Moreover, forced expression of both genes could not maintain pluripotency in primed EpiSCs (Supplementary Fig. 4) and Klf7 expression in murine ESCs had no effect[40], indicating that the two factors are human-specific pluripotency regulators.

It will be interesting to test whether the functions of TGF-beta and its direct targets are conserved or divergent in naïve and primed hPSCs.

Inhibitors of differentiation (ID) genes, such as ID1, block neural differentiation in the developing mouse embryo[51] and in murine pluripotent stem cells[52]. ID1 is induced by BMP and by TGF-beta[52,53], also shown in our experiments. ID1 expression had a mild yet reproducible effect on AP-positive colony formation and maintenance of PRDM14 (Fig. 3a), and in the future it will be interesting to study whether ID1 inhibits neural differentiation also in hPSCs.

ZNF398 is a member of the Krüppel-associated box domain zinc finger proteins (KZFPs), the largest family of transcriptional regulators found in higher vertebrates. The majority of the 350 KZFPs identified in humans have been found to be associated with repression of transposable elements[54], playing key roles during early embryogenesis. Interestingly, roughly one-third of KZFPs, were found to be associated with gene promoters, as in the case of ZNF398.

We are tempted to speculate that some members of such a large family might have acquired new roles, beyond silencing of transposable elements and in so doing, contributed to the evolution of gene-regulatory networks.

ZNF398, also known as ZER6, has never been implicated in regulation of pluripotency. Previous studies reported that ZNF398 directly activates transcription[44] and is regulated by Oestrogen Receptor Alpha[44,55]. Two isoforms of ZNF398 have been described[44,55,56], called p71 and p52. The shorter isoform (p52), lacks a N-terminal domain and promotes proliferation of cancer cells by ubiquitination of p53[56]. In hPSCs the longer isoform (p71) is predominant and has been used in all our experiments. It will be interesting to test whether p52, which lacks the N-terminal domain, regulates pluripotency in hPSCs.

Our results have also potential implications for reprogramming: ZNF398 knockdown strongly reduced reprogramming efficiency, indicating a critical role during establishment of pluripotency.

In particular, we observed reduced morphological conversion from mesenchymal to epithelial-like cells and reduced expression of epithelial markers and pluripotency markers, further indicating that ZNF398 promotes both pluripotency and epithelial character.

It will be interesting to see if ZNF398, or other members of the extended set of human pluripotency regulators, can be used to generate iPSCs at higher efficiency or to identify fully reprogrammed cells.

## Methods
**Cell culture**. hESCs (HES2, H9 and BG01V/hOG [BG01V, Gibco R7799105]) and hiPSCs (KiPS, Keratinocytes induced Pluripotent Stem Cells) were cultured in feeder-free on pre-coated plates with 0.5% growth factor-reduced Matrigel (CORNING 356231) (vol/vol in PBS with MgCl2/CaCl2, Sigma-Aldrich D8662) in E8 medium (made in-house according to Chen et al. [4]) or in mTeSR (StemCell Technologies 05850) at 37 °C, 5% CO2, 5% O2. Cells were passaged every 3–4 days at a split ratio of 1:8 following dissociation with 0.5 mM EDTA (Invitrogen AM99260G) in PBS without MgCl2/CaCl2 (Sigma-Aldrich D8662), pH8. The human foreskin fibroblasts BJ (passage 12, ATCC, CRL-2522) were cultured in DMEM/F12 (Sigma-Aldrich D6421) with 10% foetal bovine serum (FBS; Sigma-Aldrich F7524) at 37 °C, 5% CO2, 21% O2. The H9 line (WA09) was obtained from and used under authorisation from WiCell Research Institute. The KiPS line was derived by reprogramming of human keratinocytes[21] (Invitrogen) with Sendai viruses encoding for OSKM and kindly provided by Austin Smith's laboratory. The HES2 line was derived from a female human embryo at the blastocyst stage, as described in ref. [57] and kindly provided by Nicola Elvassore's laboratory.

EpiSC lines (GOF18[27] and OEC2[38], kindly provided by Hans R. Schöler's laboratory and Austin Smith's laboratory, respectively) were cultured on serum-coated (GMEM [Sigma-Aldrich G5154] with 10% FBS) plates in serum-free media N2B27 (DMEM/F12 [Gibco 11320-074], and Neurobasal in 1:1 ratio [Gibco 21103-049], with 1:200 N2 Supplement [Gibco 17502-048], and 1:100 B27 Supplement [Gibco 17504-044], 2 mM L-glutamine [Gibco 25030-024], 0.1 mM 2-mercaptoethanol [Sigma-Aldrich M3148]) supplemented with FGF2 (12 ng/ml, QKINE Qk002, recombinant zebrafish FGF2) and Activin A (20 ng/ml, QKINE Qk001), and passed as small cell clumps every 2 days.

MCF10A and MCF10neoT were cultured in DMEM/F12 with 5% horse serum (HS) (ThermoFisher 16050-122), 10 μg/ml insulin (Sigma-Aldrich I9278), 100 ng/ml cholera toxin (Sigma-Aldrich C8052), 20 ng/ml hEGF (Peprotech AF100-15), 500 ng/ml hydrocortisone (Sigma-Aldrich H0396) and 2 mM L-glutamine. RPE-1, MCF10CA1a, A549 and MDA-MB-231 were cultured in DMEM/F12 with 10% FBS and 2 mM L-glutamine. HEK293T and HaCaT were cultured in DMEM (Gibco 41965-039) with 10% FBS and 2 mM L-glutamine. WI-38 cells were cultured in MEM (Gibco 32360-026) with 10% FBS and 5% $O_2$. HepG2 were cultured in MEM with 10% FBS, 1.5% MEM non-essential amino acids (NEAA, Invitrogen 1140-036) and 4 mM L-glutamine. MCF10A, MCF10AneoT, RPE-1, MCF10CA1a, A549, MDA-MB-231, HEK293T, WI-38 and HepG2 were kindly provided by Sirio Dupont's laboratory. HaCaT cells were kindly provided by Stefano Piccolo's laboratory.

All cell lines were mycoplasma-negative (Mycoalert, Lonza).

**Treatment with inhibitors and cytokines.** Treatments were performed either under feeder-free conditions or on feeders (MEF, Murine Embryonic Fibroblasts mitotically inactivated, DR4 ATCC). For the validation experiments of Fig. 2a in feeder-free, KiPS were plated on plastic coated with 0.5% Matrigel. The next day, cells were treated with DMSO (Sigma-Aldrich D2650) or 10 μM SB43 (Axon Medchem 1661) overnight. The morning after, TGF-beta signalling was re-induced by changing medium with mTeSR1 for 1 h or for 4 h. For the validation experiments on feeders, KiPS were plated on MEF with KSR medium [DMEM/F12, with 20% KnockOut Serum Replacement (KSR, Gibco 108828028), 2 mM L-glutamine, 1% NEAA and 0.1 mM 2-mercaptoethanol] and with 10 ng/ml FGF2. The next day, cells were treated with DMSO or with 10 μM SB43 overnight. The morning after, cells were treated with 2 ng/ml of TGFB1 (Peprotech 100-21) or with 25 ng/ml of Activin A.

For the BMP induction experiment in Supplementary Fig. 6c, KiPS were plated under feeder-free conditions. The next day, cells were treated with DMSO or 0.1 μM LDN 193189 (LDN, Axon Medchem 1509) overnight. The morning after, BMP signalling was re-induced by changing medium with E8 with 100 ng/ml of BMP4 (Peprotech 120-05ET) for 1 h.

**Generation of hPSCs stably expressing genes of interest.** Stable transgenic hPSCs expressing candidates were generated by transfecting cells with PB transposon plasmids with PB transposase expression vector pBase. In order to generate the PB plasmids, the candidates (NANOG, ZNF398, KLF7, MYC, ETS2, OTX2, ID1, BCOR and PRDM14) were amplified from cDNA and cloned into a pENTR2B donor vector. Then, the transgenes were Gateway cloned into the same destination vector containing PB-CAG-DEST-bghpA and pGK-Hygro selection cassette.

For DNA transfection, 250,000 hPSCs were dissociated as single cells with TrypLE (Gibco 12563-029) and were co-transfected with PB constructs (550 ng) and pBase plasmid (550 ng) using FuGENE HD Transfection (Promega E2311), following the protocol for reverse transfection. For one well of a 12-well plate, we used 3.9 μl of transfection reagent, 1 μg of plasmid DNA, and 250,000 cells in 1 ml of E8 medium with 10 μM Y27632 (ROCKi, Rho-associated kinase (ROCK) inhibitor, Axon Medchem 1683). The medium was changed after overnight incubation and Hygromycin B (200 μg/ml; Invitrogen 10687010) was added after 48 h. For the overexpression experiments, hPSCs stably expressing an empty vector or the candidates were plated. The next day, cells were treated with DMSO or 10 μM SB43 for 5 days and then analysed as indicated in Supplementary Fig. 2a.

**Murine EpiSCs experiments.** For generation of stable transgenic lines over-expressing candidate genes, EpiSCs were reverse-transfected with 3 μl of Lipofectamine 2000 (Invitrogen 11668-019) using 500 ng of PB transposon plasmid harbouring the indicated factor and 500 ng of transposase in Opti-MEM (Gibco 51985-026). $1.2 \times 10^5$ cells in 800 μl N2B27 with FGF2 (12 ng/ml) and Activin A (20 ng/ml) and 10 μM ROCKi were added to the transfection mix and plated in serum-coated 12-well plates. The next day the medium was changed and Hygromycin B selection was applied for 5 days. To test the effect of TGF-beta inhibitors, 1/20 of a confluent well was plated on serum-coated 12-well in N2B27 medium with FGF2 and Activin A. The next day, the medium was changed to N2B27 with FGF2 and 1 μM SB43 or FGF2 and 1 μM A83 (Axon Medchem 1421). After 48 h, cells were harvested for expression analysis.

**siRNA and DNA transfection in HEK293T and HaCaT cell lines.** Cells were plated at 20% confluence on a 24-well plate the day before transfection. For transfection with siRNAs, each individual well was transfected with Lipofectamine RNAiMAX reagent (ThermoFisher 13778075) following the manufacturer protocol (0.2 μl of 100 μM siRNA with 1 μl of transfection reagent per well). For transfection with DNA, each individual well was transfected with a mix of: 2.25 μl of poly-ethylenimine (PEI, Polysciences 23966), 750 ng DNA in 100 μl Opti-MEM. In cases of treatment with TGF-beta, cells were starved in medium without serum (+10 μM SB43, for SB43 samples only), 24 h after transfection. After overnight incubation, the medium was replaced with DMEM without serum and with 10 μM SB43 or 5 ng/ml TGFB1 for 6 h.

**siRNA transfection in hPSCs.** For siRNA transfection, hPSCs were plated on Matrigel-coated 24-well plate as clusters (2500–5000 clusters for one well of a 24-well plate) in E8 medium with 10 μM ROCKi. After 4 h, siRNAs were transfected at a final concentration of 20 nM using Stemfect$^{TM}$ RNA Transfection Kit (STEMGENT 00-0069), following the protocol for forward transfection.

For a 24-well plate (2 cm$^2$), we used 0.52 μl of transfection reagent, 2 μl of 10 μM siRNA solution and 25 μl of transfection buffer. After waiting 20 min, we mixed the transfection mix with 1 ml of E8 medium. The medium was changed after overnight incubation. See Supplementary Table 1 for sequences of the siRNAs used.

**EBs differentiation assay.** KiPS stably expressing an empty vector or ZNF398 were detached as clumps with EDTA and plated on ultra low attachment surface plates (CORNING 3473) in E8 medium with 10 μM ROCKi. After 2 days, E8 medium was substituted with DMEM, 20% FBS, 2 mM L-glutamine, 1% NEAA and 0.1 mM 2-mercaptoethanol. Medium was changed every 2 days.

**Reprogramming.** All reprogramming experiments were performed in micro-fluidics in hypoxia conditions (37 °C, 5% $CO_2$, 5% $O_2$)[48]. The protocol for reprogramming experiments was optimised to transfect siRNA in order to test the requirement of ZNF398 for reprogramming.

Briefly, microfluidic channels were coated with 25 μg/ml Vitronectin (ThermoFisher, A14700) for 1 h at room temperature (RT). In the case of OSKMNL reprogramming, fibroblasts were seeded at day 0 at 30 cells/mm$^2$ in DMEM/10% FBS. On day 1, 9 h before the first mRNAs transfection, we applied E6 medium (made in-house according to Chen et al.[4]) including 100 ng/ml FGF2, 5 μM ROCKi, 0.1 μM LSD1i (RN-1, EMD Millipore 489479) and 20% KSR (Gibco, 10828028). The transfection mix was prepared according to the StemMACS$^{TM}$ mRNA Transfection Kit (Miltenyi Biotec, 130-104-463) and Stemgent StemRNA-NM Reprogramming Kit (Reprocell, 00-0076) (OSKMNL not-modified RNA (NM-RNA) and EKB NM-RNA (used to reduce interferon response) and we prepared the RNA mix according to the manufacturer's instructions.

In the case of OSKM reprogramming, individual modified mRNAs (OCT4, SOX2, KLF4 and MYC) were made in-house by in vitro transcription using mRNA synthesis with HiScribe$^{TM}$ T7 ARCA mRNA Kit (NEB E2060S) according to the manufacturer's instructions. On day 0, fibroblasts were seeded at 15 cells/mm$^2$ in DMEM/10% FBS. On day 1, 9 h before the first mRNAs transfection, we applied E6 medium including 100 ng/ml FGF2, 5 μM ROCKi, 0.1 μM LSD1i, 1% KSR and 200 ng/ml B18R (Invitrogen 34-8185-81). The B18R protein was added to the medium to reduce the interferon response. The transfection mix was prepared according to the StemMACS$^{TM}$ mRNA Transfection Kit and using OSKM mRNAs made in-house and NM-microRNAs (Stemgent StemRNA-NM Reprogramming Kit). Cells were transfected daily at 6 p.m. and fresh medium was given daily at 9 a.m. siRNAs were transfected at a final concentration of 20 nM at day 1, day 3 and day 5 (see Supplementary Table 1 for sequences of the siRNAs used) together with mRNAs. The dose of mRNAs transfected was gradually increased according to cell proliferation rate and transfection-induced cell mortality[48].

**Immunofluorescence and stainings.** Immunofluorescence analysis was performed on 1% Matrigel-coated glass coverslip in wells or in situ in microfluidic channels with the same protocol. Cells were fixed in 4% formaldehyde (Sigma-Aldrich 78775) in PBS for 10 min at RT, washed in PBS, permeabilized for 1 h in PBS + 0.3% Triton X-100 (PBST) at RT, and blocked in PBST + 5% of HS (ThermoFisher 16050-122) for 5 h at RT. Cells were incubated overnight at 4 °C with primary antibodies (see Supplementary Table 2) in PBST + 3% of HS. After washing with PBS, cells were incubated with secondary antibodies (Alexa, Life Technologies) (Supplementary Table 2) for 45 min at RT. Nuclei were stained with either DAPI (4′,6-diamidino-2-phenylindole, Sigma-Aldrich F6057) or Hoechst 33342 (ThermoFisher 62249). In the case of Phalloidin staining (see Fig. 8f), Alexa Fluor 488 Phalloidin and Hoechst were added with secondary antibodies. Images were acquired with a Zeiss LSN700 or a Leica SP5 confocal microscope using ZEN 2012 or Leica TCS SP5 LAS AF (v2.7.3.9723) software, respectively.

For alkaline phosphatase staining, cells were fixed with a citrate–acetone–formaldehyde solution and stained using an alkaline phosphatase detection kit (Sigma-Aldrich 86R-1KT). Plates were scanned using an Epson scanner and scored manually.

**Image analysis.** Fiji 1.0 (ImageJ2)[58] was used for image analysis. Fluorescence intensity across hPSCs (Supplementary Fig. 6b) was measured using the Plot Profile function. For each condition, 48 cells from six randomly selected fields were analysed. Fluorescence intensity (Fig. 4a, Supplementary Fig. 3c) was quantified using Cell Profiler software (v3.1.8).

**Western blotting.** To monitor endogenous protein levels, cells were detached, medium removed and frozen at −80 °C prior to processing. Pellets were then thawed and resuspended in 10 ml/cm$^2$ HPO buffer (50 mM Hepes pH 7.5, 100 mM NaCl, 50 mM KCl, 1% triton X-100, 0.5% NP-40, 5% glycerol, 2 mM $MgCl_2$) freshly supplemented with 1 mM DTT, protease inhibitors (Roche 39802300) and phosphatase inhibitors (Sigma-Aldrich P5726). Western blotting was performed as in ref.[59].

Western blotting was acquired with LAS400 ImageQuant 1.2. Antibodies are detailed in Supplementary Table 2. Uncropped gels are provided in the Source data file.

**Quantitative PCR.** Total RNA was isolated using Total RNA Purification Kit (Norgen Biotek 37500), and complementary DNA (cDNA) was made from 500 ng using M-MLV reverse transcriptase (Invitrogen 28025-013) and dN6 primers. For real-time PCR SYBR Green Master mix (Bioline BIO-94020) was used. Primers are detailed in Supplementary Table 3. Three technical replicates were carried out for all quantitative PCR. GAPDH was used as endogenous control to normalise expression. qPCR data were acquired with QuantStudio™ 6&7 Flex Software 1.0.

**RNA sequencing.** For induction experiments (Fig. 1d), poly(A) mRNA was purified from total RNA using the Dynabeads mRNA direct kit (ThermoFisher, 61011). Quantity and quality of the starting mRNA were checked by Qubit and Agilent Bioanalyzer 2100 RNA pico chip. The template library was prepared using the Ion Total RNA-Seq Kit v2 (ThermoFisher, 4475936). Quantity and size distribution of the library were analysed using the Agilent Bioanalyzer 2100 DNA HS chip. Emulsion PCR using 10 ml of 100 pM library was performed using a One-Touch 2 instrument (ThermoFisher, 4474778) with an Ion PI Template OT2 200 kit following the manufacturer's instructions (ThermoFisher, 4488318). The enrichment of the template library was achieved using the Ion OneTouch ES enrichment system (ThermoFisher). Ion Proton sequencer and IPv2 chip were prepared according to the manufacturer's recommendations. Raw reads were aligned in two steps: first reads were aligned on genome build GRCh37.p13 with STAR (v2.4), reads that were not aligned in this step were realigned with bowtie2 (v2.2.4). Raw counts over the ensembl annotation release 75 were obtained with htseq-count (v0.6.0). Normalisation and differential analysis were carried out using edgeR package (v3.4.2)[60] and R (version 3.5.2, R Core Team (2018). R: A language and environment for statistical computing. R Foundation for Statistical Computing, Vienna, Austria. https://www.R-project.org/). EdgeR fits genewise negative binomial generalised linear models and conducts likelihood ratio test. Raw reads were normalised to obtain counts per million-mapped reads (CPM) and reads per kilobase per million mapped reads (RPKM). Only genes with a CPM >1 in at least two samples were retained for differential analysis. Differences between batches were adjusted using an additive model. Genes were considered significantly upregulated with a p-value ≤ 0.05 and a fold-change ≥1.5.

For overexpression experiments (Figs. 4b, c and 5a), ~2 μg of total RNA were subjected to poly(A) selection, and libraries were prepared using the TruSeq RNA Sample Prep Kit (Illumina) following the manufacturer's instructions. Sequencing was performed on the Illumina NextSeq 500 platform. Reads were mapped to the *Homo sapiens* hg19 reference assembly using TopHat (v2.1.1), and gene counts were computed using htseq-count (v0.6.1p1)[61]. Differential expression analysis was performed using DESeq v2[62]. Genes with abs (Log2 fold-change) ≥ 1 and p-value < 0.01 were considered significant and defined as differentially expressed (differentially expressed genes (DEG)).

GO terms for biological processes analysis of DEGs was performed using Database for Annotation, Visualisation and Integrated Discovery (DAVID) database[63] (https://david.ncifcrf.gov). Boxplots and Scatterplots were made using TPM values exploiting ggpubr R package (v. 0.2), ggboxplot and ggscatter functions, respectively. Heatmaps were produced using TPM values with the pheatmap function from pheatmap R package (v.1.0.12, distance = 'correlation', scale = 'row') on selected markers. Volcano plots were computed with Log2 fold-change and −Log10 p-value using ggscatter function from ggpubr R package (v. 0.2).

**Microarray analysis.** Public gene expression data of hESCs treated with SB43 were downloaded from ArrayExpress (E-MEXP-1741). Differentially expressed genes were identified applying limma (v3.18.13)[64] on the RMA normalised gene expression matrix. Limma fits a linear model for each gene and calculates moderated t-statistics and p-values with an empirical Bayes moderation approach.

To identify genes associated with TGF-beta inhibition, we compared the expression levels of hESCs treated with SB43 or control cells and selected those probe sets with a fold change lower than or equal to −2 and an FDR lower than or equal to 0.05. Microarray analyses were performed in R (version 3.5.2).

**ChIP sequencing and ChIP quantitative PCR.** ChIP-seq data of SMAD3 in BG03 embryonic stem cells were retrieved from GEO (GSE21614). We analysed the chromatin IP against Smad3 (GSM539548) and whole cell extract (WCE) in the same cell line (GSM539552). Raw reads were aligned using Bowtie (version 0.12.7)[65]; to build version hg19 of the human genome retaining only uniquely mapped reads. Redundant reads were removed using SAMtools (v0.1.18). MACS2 (v2.0.10)[66] was used to call peaks for SMAD3 using WCE ChIP-seq as control sample and setting the bandwidth equal to the estimated sonication fragment size (131 bp) and the p-value cutoff at 0.01. Only peaks with a pileup height >5 were kept for further analysis. Each peak was assigned to the nearest TSS in a window of 100 kb centred on the peak, considering only protein-coding genes in GENCODE v16 annotation.

For identification of ZNF398 targets, we performed chromatin immunoprecipitation in two independent hESC lines (H9 and BG01V)[67,68]. Cells (~3 × 10[7]) co-transfected with Avi-Tag-ZNF398 and *E. coli* birA protein were

crosslinked in 1% formaldehyde for 10 min at room temperature. Crosslinking was quenched by addition of 0.125 M final glycine. Cells were then harvested by scraping in ice-cold PBS and collected by centrifugation. The cell pellet was then resuspended in 1 ml ice-cold ChIP buffer [20 mM Tris–HCl pH 8.0; 0.1% SDS; 1% Triton X-100; 2 mM EDTA; 150 mM NaCl], supplemented with protease inhibitor cocktail (Sigma-Aldrich, P8340) and incubated on ice for 10 min. The cell suspension was then sonicated with a Diagenode Bioruptor Twin (settings: 30 s ON, 30 s OFF, high power) for 10 cycles. The sample was then kept on ice for 10 min and sonication was repeated for additional 10 cycles. The lysate was then centrifuged at 17,000×g for 10 min (4 °C) to remove membranes and the supernatant was transferred to a new tube. 50 μl of Dynabeads MyOne Streptavidin T1 (Thermo Fisher, 65601), pre-equilibrated for 30 min in PBS supplemented with 1% BSA, were then added to the sample. The sample-beads suspension was then rotated at 4 °C for 3 h. Following incubation, supernatant was discarded and beads were washed (in 1 ml volume) twice with Wash buffer 1 [2% SDS], twice with Wash buffer 2 [50 mM HEPES pH 7.5; 500 mM NaCl; 1 mM EDTA; 1% Triton X-100; 0.1% sodium deoxycholate], once with Wash buffer 3 [10 mM Tris–HCl pH 8.0; 250 mM LiCl; 1 mM EDTA; 0.5% NP-40; 0.5% sodium deoxycholate] and once in TE buffer. Beads were then resuspended in 200 μl Elution buffer [50 mM Tris–HCl pH 8.0; 10 mM EDTA; 1% SDS] and incubated at 56 °C for 16 h. After incubation, beads were discarded and five volumes (1 ml) of buffer PB were added to the supernatant, prior to DNA purification on QIAquick PCR Purification kit's columns (QIAGEN, 28104), according to the manufacturer instructions. The ChIP-seq library was prepared with ~5 ng of immunoprecipitated DNA as input for the NEBNext® ChIP-Seq Library Prep kit, following the manufacturer's instructions. Sequencing was performed on the Illumina NextSeq 500 platform. Reads were mapped to the *Homo sapiens* hg19 reference assembly using Bowtie (v1.2.2)[65], keeping only uniquely mapped reads. Reads (75 bp) were bioinformatically extended to the average insert size (150 bp), and identical reads (reads starting and ending at the same positions) were collapsed. Peak calling was performed using MACS v2.1.1[66], selecting only peaks with q-value < 0.05. A non-redundant set of common peaks between the two ZNF398 ChIP-seq replicates was generated using the *intersectBed* utility from BEDTools (v2.26.0)[69]. For motif discovery, peaks were resized to ±200 bp surrounding their center and motif discovery was performed using MEME (v4.10.1)[70]. For correlation analyses and comparison of ZNF398 genome occupancy with known factors/histone modifications, data was collected from the GEO database for the following datasets: GSE54471 (H3K27ac and H3K4me1), GSE76084 (H3K27me3, H3K36me3, H3K4me3, H3K9ac, SOX2), GSE118325 (H3K9me3), GSE73725 (NANOG). Data for POU5F1 and EP300 was instead obtained from the ENCODE database (https://www.encodeproject.org/). All samples were analysed as stated above. Spearman correlations between genomic occupancy profiles were computed using the *multiBamSummary* and *plotCorrelation* utilities from deepTools v2.2.4[71]. Heatmaps of peak densities around ZNF398 peaks centers were generated using in-house developed scripts.

For SMAD3 and SMAD2 ChIP-qPCR, hESC lines (H9 and BG01V) were treated with 25 ng/ml Activin A to activate the TGF-β pathway for 1 h and cross-linked by addition of formaldehyde to 1% for 10 min at RT, quenched with 0.125 M glycine for 5 min at RT, and then washed twice with cold PBS. The cells were resuspended in Isotonic buffer supplemented with 1% NP-40 to isolate nuclei. The pellets were then resuspended in ChIP buffer (20 mM Tris–HCl pH 8.0, 10 mM EDTA, 1% SDS). Extracts were sonicated using the BioruptorH Twin (Diagenode) for two runs of 10 cycles (30 s on, 30 s off) and diluted with ChIP dilution buffer (20 mM Tris–HCl pH 8.0, 150 mM NaCl, 2 mM EDTA, 1% Triton) before the immunoprecipitation step with 2 μg of antibody overnight at 4 °C on a rotator. Subsequently immunoprecipitated complexes were washed six times with RIPA buffer (50 mM HEPES–KOH pH 7.6, 500 mM LiCl, 1 mM EDTA, 1% NP-40, 0.7% Na-Deoxycholate) and eluted in SDS Elution buffer. De-crosslinked DNA was purified using QiaQuick PCR Purification Kit (Quiagen) according to the manufacturer's instruction.

The ChIP-seq data were validated by ChIP–qPCR, using two independent biological replicates for each hESC lines (H9 and BG01V). The data represent qPCR measurements of the immunoprecipitated DNA performed using SYBR GreenER kit (Invitrogen) and were normalised to those obtained with a non-immune serum (IgG). The data are expressed as a percentage of the DNA inputs. Primers for ChIP–qPCR are detailed in Supplementary Table 4.

**Protein coimmunoprecipitation.** To detect the protein interaction, nuclei were isolated from H9 cells expressing Avi-Tag-ZNF398 which were induced with 25 ng/ml Activin A for 1 h. Cells were lysed with Isotonic buffer supplemented with 1% NP-40. The nuclei pellets were resuspended in IP buffer (50 mM Tris–HCl pH 8.0, 100 mM NaCl, 200 mM sucrose, 0.5 mM MgCl₂, 5 mM CaCl₂, 5 μM ZnCl₂) and were treated with micrococcal nuclease at 30 °C for 10 min. Nuclear proteins were incubated with 2 μg of indicated antibodies (Supplementary Table 2) overnight at 4 °C. The immunoprecipitated complexes were incubated with Protein G magnetic beads (Invitrogen) for 2 h at 4 °C and then were washed three times with IP buffer plus 0.5% NP-40. The precipitated proteins were eluted by incubating with 0.5 M NaCl TE buffer and were further analysed by western blotting.

**Statistics and reproducibility.** For each dataset, sample size *n* refers to the number of independent experiments or biological replicates, shown as dots, as

stated in the figure legends. A Gaussian distribution was not assumed and p-values were calculated using the non-parametric unpaired two-tailed Mann-Whitney U test with the exception of induction experiments (Fig. 2a, b) for which we used the unpaired two-tailed t-test. p-values were not calculated for datasets with n < 3.

p-values are reported in the plots or figure legends. R software (v3.5.2) was used for statistical analysis.

All error bars indicate the standard error of the mean (SEM). All key experiments were repeated between two and five times independently, as indicated. Experiments of candidate's functional validation were repeated using three different hPSC lines. All qPCR experiments were performed with three technical replicates.

**Reporting summary**. Further information on research design is available in the Nature Research Reporting Summary linked to this article.

## Data availability

RNA-seq and ChIP-seq data for this study have been deposited in the Gene Expression Omnibus (GEO) database under the accession code: GSE133630 . For the identification of TGF-beta transcriptional targets, we used available SMAD3 ChIP-seq data from[15] (Accession no. GSE21621), microarray data from[13] (Accession no. E-MEXP-1741) and RNA-seq data of H9 from[72], (Accession no. GSE24447, see Supplementary Fig. 1e). For correlation analyses and comparison of ZNF398 genome occupancy with known factors/histone modifications, data was collected from the GEO database for the following datasets: GSE54471 (H3K27ac and H3K4me1), GSE76084 (H3K27me3, H3K36me3, H3K4me3, H3K9ac, SOX2), GSE118325 (H3K9me3), GSE73725 (NANOG). Data for POU5F1 and EP300 was instead obtained from the ENCODE database (https://www.encodeproject.org/). All plasmids, materials and data supporting the findings of this study are available from corresponding authors upon reasonable request. The source data underlying Figs. 1b, 2a, b, 3a, c, 4a, 5d, e, 6c, 7a, b, 8c, e and Supplementary Figs. 1b, f, g, 2d, 3a, c, d, 4a, b, 5a–c, 6b, 7a-d, 8a, c–e are provided as a Source Data file.

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

## Acknowledgements

The authors thank S. Dupont, A. Ditadi and S.J. Dunn for critical reading of the manuscript, and the Martello Laboratory for discussions and suggestions. G.M.'s Laboratory is supported by grants from the Giovanni Armenise–Harvard Foundation, the Telethon Foundation (TCP13013) and an ERC Starting Grant (MetEpiStem). S.O. Laboratory is supported by grants from Associazione Italiana Ricerca sul Cancro (AIRC-IG 2017 Id. 20240) and PRIN 2015. We also thank the Italian Epigenomics Flagship Project (Epigen) for supporting M.F. and G.M.T.

## Author contributions

G.M. and S.O. designed the study; I.Z. and M.P. performed all functional experiments in hPSCs and reprogramming assays; I.Z., M.P. and M.A. performed molecular characterisation of hPSCs; M.F. and G.M.T. performed SMAD3 targets identification; D.I. performed ChIP-seq and RNA-seq experiments and analyses; M.M. performed ChIP-qPCR and Co-IP experiments; M.A. performed all additional bioinformatic analyses; E.C. performed all EpiSCs experiments, Marco Montagner performed experiments with human cancer cell lines; M.A., I.Z. and D.I. prepared the figures; G.M. wrote the manuscript with input from all authors; G.M. and S.O. supervised the study and provided fundings.

## Competing interests

The authors declare no competing interests.
