## [Peer Review File · Nature Communications]

Reviewers' Comments:

Reviewer #1:

Remarks to the Author:

Zorzan and co-workers explored the role of TGFbeta signaling in pluripotency and the maintenance/proliferation of human pluripotent stem cells. The aim of this study is to identify transcription factors induced by TGFbeta controlling pluripotency. They focus on Smad3 binding factors and identified ZNF398 to be important for TGFbeta mediated pluripotency.

This is an interesting study providing new insight but there there are some issues that need to be resolved.

- 1) Why do the authors focus on Smad3 and do not involve Smad2 in their investigations?
- 2) SB43 is not a specific TGFb inhibitor but will inhibit ALK4, ALK5 and ALK7 kinase activity. This should be taken into consideration.
- 3) TGFbeta signaling uses different combinations of type I, type II and type III receptors. Depending on the ligand available (TGFbeta/Activin) and the receptor combination, different effects are possible. No information is given on the receptors present on these cells nor on the induction of pSmads upon ligand addition.
- 4) In most cell types, TGFbeta induces EMT in collaboration with Smad3. Why is this different in these cells and is ZNF398 unique for hPSC? Is the same effect seen in other epithelial cells types?
- 5) TGFb and activin have different responses in the presence or absence of feeders. The effect seen is more dominant when no feeders are used. Is this related to different media used? Most of the serum free / chemical defined media contain TGFb which might be at different levels in the feeder cultures.
- 6) They identify ID1 as a bonafide TGFbeta target, but this has been reported as a BMP/Smad1/5 target. Please explain.
- 7) They use AP activity as a readout for pluripotency. In the presence SB, so lack of TGFb and related to more BMP signaling, the authors report more AP activity. AP is also a marker for osteogenic differentiation, which is induced by BMP. How are the authors able to discriminate between the two?
- 8) In figure 3 they find genes related to mesenchymal transition and extracellular matrix organization, however this is normally related to TGFbeta signaling. In this study it is related to inhibition of TGFb signaling. Please explain.
- 9) In figure 1A, please show the expression levels/localization of pSmad2, 3 and pSmad1 as inhibiting TGFbeta signaling often enhances BMP signaling, also important for pluripotency.
- 10) Why not feeder-free and activin? What receptors are expressed?
- 11) Although significant, is the induction of Nanog and ZNF398 also relevant? Because fold of induction is shown, it is not clear what the levels are. Smad7, as a direct TGFb target, is much more induced. Is this a more relevant gene?

Minor

Line 76 and to should be and the
Line 85 the down regulated genes

Reviewer #2:

Remarks to the Author:

In this manuscript, the authors reported the identification of the genes responsible for mediating the action of the TGFb signal in human pluripotent stem cells (hPSCs). The Activin/TGFb signal is essential for self-renewal of hPSCs in conventional culture condition but the molecular mechanism to mediate the action of this signal is largely unknown. The authors applied a sophisticated way of the analysis to identify the functional targets of the TFGb signal in hPSCs. As the result, they identified 4 genes encoding transcription factors (NANOG, MYC, KLF7 and ZNF398) that are

regulated by the TGF β signal and sufficient to replace its function by their over-expression. Among them, the authors chose ZNF398 for further analysis. They found that ZNF398 co-occupy the target sites in the genome with SMAD3 and EP300. Moreover, they demonstrated that ZNF398 is essential for efficient reprogramming of the somatic cells to hPSCs by the OSKMNL factors. These results emphasize the role of ZNF398 to maintain and establish pluripotency.

ZNF398 is a mysterious factor since there is no previous report for its function in PSCs in any species. In somatic cells, it was recently reported that p52-ZER6 (encoded by ZNF398) interacts with TRP53 and modulates its stability (Huang et al, EBioMedicine, 2019: PMID 31521611). According to HPA RNA-seq data in normal tissues, the expression of ZNF398 is quite homogeneous, suggesting its general function. Therefore, the inefficient reprogramming by the knockdown of ZNF398 could not solely due to the failure of the activation of pluripotency and MET-associated genes. The co-localization with SMAD3 may suggest its role to mediate the TGF β signal in any cell types, not specific to hPSCs. This point should be addressed carefully in the revised manuscript. In addition, there are several points required revision for publication in Nat Commun.

1. In Fig 2, the authors showed the impact of the overexpression of 4TFs they identified to maintain self-renewal in the absence of the TGF β signal. However, they look just short period of culture (for 5 days in the presence of SB43). How about their powers in longer culture period with passages in SB43?
2. Line 141: The authors stated the similarity between KLF7 and KLF2/4/5, but their distinct function to maintain pluripotency was demonstrated previously (Yamane et al, Development, 2018).
3. As mentioned above, ZNF398 could be a general co-factor of SMAD3 to support its function. The authors should address this possibility in appropriate somatic cell line that respond to the TGF β signal and express ZNF398.
4. Line 270: Is Dppa4 a human-specific pluripotency factor? Recent reports indicated that Dppa2/4 involve in the activation of 2C genes in mouse ES cells (PMID: 30948459, 30692203).
5. It will be nice if the authors show the function of the 4 TFs in mouse epiblast stem cells to replace the action of the TGF β signal.

Reviewer #3:

Remarks to the Author:

Irene Zorzan and colleagues performed a global analysis of the transcriptional program controlled by TGF beta followed by an unbiased gain-of-function screening in multiple hPSC lines to identify factors mediating TGF beta activity. They found a new downstream mediator of TGF beta signal, ZNF398, which could induce genes associated both with pluripotency and epithelial character together with SMAD3 and the histone acetyltransferase EP300. This is an interesting work on a relatively poorly explore field, as it is human pluripotency, and it, thus, has potential relevant implications. In general, I liked the paper but before publication in Nature Communications there are several aspects that should be improved, in my opinion.

Major and minor problems:

1. In line 99 the authors mention 'extensive validation' but something relevant like a ChIP-qPCR panel is missing. Related, the plot in Figure 1e is not immediately intuitive.
2. The evidence for co-localization and functional interaction of ZNF398 with SMAD3 and EP300 is not enough. The authors could do co-immunoprecipitation and/or ChIP-on-CHIP.
3. Related to the previous item. In lines 205-207, the authors wrote "We extended our analysis to all 81 SMAD3 direct target genes (Fig. 1c) and observed that 28.4% of them were also co-bound by ZNF398 (enrichment of 3.67 fold over expected, p-value = 3.49e-12, Chi-squared test)". Please

put this on a Figure to support the observation.

4. The cut-offs in for example Figures 3a and 4f, aren't they too low?

5. ChIP-qPCR showing binding of ZNF398 to some target genes is desirable.

6. The effect on epithelial genes in reprogramming after knockdown of ZNF398 is modest (e.g., Figures 5e and d). Since the authors reprogram the fibroblasts with redundant genes (according to their screen), wouldn't it be better to reprogram with less factors (e.g., the 4 Yamanaka factors) to see a more potent effect of the ZNF398 knockdown. Related, the authors measure colony formation at day 14, which is not representative of reprogramming efficiency but rather the mesenchymal-to-epithelial transition, so please modify the statements.

7. Many references should be added to refer to the work of others. The figures should be mentioned in order to avoid confusion; Supplementary Figure 3c wasn't mentioned in the text. For Figure 4c, a different set of genes are mentioned in the text. Add error bars to Supplementary Figure 2b. Mention clearer how many times was each experiment repeated. Is Figure 3a first mentioned correctly in lines 134-136?

8. The methods need to be revised and explained in more detail. For example: why in lines 577 and 589 the authors use different tools? why the authors use a relatively old version of R (3.0) rather than an updated one? is the cut off in line 583 similar to the one used in the figures?

9. Some parts of the text can be written clearer, for example when referring to TGF beta-like cytokines and receptors.

Referee 1

Zorzan and co-workers explored the role of TGF-beta signaling in pluripotency and the maintenance/proliferation of human pluripotent stem cells. The aim of this study is to identify transcription factors induced by TGFbeta controlling pluripotency. They focus on Smad3 binding factors and identified ZNF398 to be important for TGF-beta mediated pluripotency.

This is an interesting study providing new insight but there are some issues that need to be resolved.

We thank the Referee for finding our study interesting and for the constructive comments.

1) Why do the authors focus on Smad3 and do not involve Smad2 in their investigations?

We focused only on SMAD3 for our candidate identification for several reasons.

We have measured the relative abundance of SMAD2 and SMAD3 in hESCs, using an antibody recognizing both proteins and found that SMAD3 is much more abundant than SMAD2 under the conditions we used (E8 or mTeSR on Matrigel). We have now included such results in Supplementary Fig. 1d.

It is also known that SMAD2 and SMAD3 form either homodimers or heterodimers (Lucarelli et al., 2018), indicating that they share the same targets. Indeed, it was recently shown that Smad2 and Smad3 have redundant functions in pluripotent cells (Senft et al., 2018). In line with this, we found that the SMAD7, NANOG and ZNF398 loci bound by SMAD3 were also occupied by SMAD2 (Fig. 1f).

Finally, it has been shown that only SMAD3 binds directly the DNA (Dennler et al., 1998, 1999; Ross and Hill, 2008), therefore we used SMAD3 ChIP-seq data for direct targets identification.

2) SB43 is not a specific TGFb inhibitor but will inhibit ALK4, ALK5 and ALK7 kinase activity. This should be taken into consideration.

We thank the Referee for pointing this out. TGF-beta 1, Nodal and Activin A are used for expansion of hESCs (Chen et al., 2011, 2006; Johansson and Wiles, 1995; Ludwig et al., 2006a, 2006b; Vallier, 2005) and both TGF-beta 1 and Activin A are able to activate the novel

pluripotency regulators we have identified. Given that they signal via ALK5 and ALK4/7, respectively, it is in fact advantageous for us to be able to block all 3 ALK proteins at the same time. In this way we are able to block completely pluripotency-supporting signals with a single inhibitor.

We have modified the text and included Supplementary Fig. 1a and 1f in order to make more clear what the inhibitor does and what receptors are expressed in hPSCs.

3) TGFbeta signaling uses different combinations of type I, type II and type III receptors. Depending on the ligand available (TGFbeta/Activin) and the receptor combination, different effects are possible. No information is given on the receptors present on these cells nor on the induction of pSmads upon ligand addition.

As mentioned above, we have now included information about which receptors are expressed in hPSCs (Supplementary Fig. 1a and Supplementary Fig. 1f). We have also performed immunostaining and Western-blot showing loss of pSMAD3 signal upon SB43 treatment and robust phosphorylation of SMAD3 upon stimulation with TGF-beta 1 for 60 minutes (Fig. 1a, and Supplementary Fig. 6b).

4) In most cell types, TGF-beta induces EMT in collaboration with Smad3. Why is this different in these cells and is ZNF398 unique for hPSCs? Is the same effect seen in other epithelial cell types?

We agree on the fact that TGF-beta induces EMT in cancer cells. TGF-beta is able to induce either growth-arrest in benign cells or to support cell proliferation in cancer cells (as in hESCs). These opposite functions of TGF-beta on different cell types is well-known and has been described as the TGF-beta paradox (Zhang et al., 2014) and partly explained by an interplay with ERK or PP2A. Thus, TGF-beta has distinct activities in a context-dependent fashion as additional signals might change the biological outcome of TGF-beta. For example it has been shown that Activin A promotes differentiation or self-renewal according to AKT signalling intensity (Singh et al., 2012). In the future, it will be interesting to study whether the proliferation and pro-Epithelial activity of TGF-beta in hESCs is regulated by other molecules, such as ERK, PP2A, and AKT.

Concerning the uniqueness of ZNF398 we have analysed transcriptomic data from a panel of healthy tissues and from human cell lines. We observed very low ZNF398 expression in normal tissues and in the majority of cell lines tested. In contrast, the known pluripotency factor KLF4

was expressed at very high levels in multiple tissues and cell lines (Supplementary Fig. 7a). We conclude that ZNF398 is highly expressed in hESCs and hiPSCs, not in normal tissues, and only in a fraction of tested human cell lines.

We also investigated the function of ZNF398 in the two cell lines (HEK293T and HaCaT, that are immortalised keratinocytes with epithelial properties). According to our model, for ZNF398 to regulate EMT it should first of all be able to regulate TGF-beta downstream genes. Upon ZNF398 knockdown or over-expression we were unable to detect any difference in the expression of the TGF-beta targets SMAD7 and JUNB (Supplementary Fig. 7b, c).

5) TGFb and activin have different responses in the presence or absence of feeders. The effect seen is more dominant when no feeders are used. Is this related to different media used? Most of the serum free/chemical defined media contain TGFb which might be at different levels in the feeder cultures.

Prompted by Referee's comment we have compared the expression of LEFTY1, a TGF-beta and Activin A target, in the presence and in the absence of feeders. As shown in Supplementary Fig. 1g, there seems to be a slightly better induction of target genes in the presence of feeders, which we speculate might be due to the production of ligands/cofactors by the feeders.

6) They identify ID1 as a bonafide TGFbeta target, but this has been reported as a BMP/Smad1/5 target. Please explain.

We thank the Referee for pointing out that ID1 is a BMP target. We have added references supporting this notion. However, ID1 had been also identified as a TGF-beta target by Liang and colleagues (Liang et al., 2009). We conclude that ID1 might be regulated by both TGF-beta and BMP. Of note, Liang reported a direct and transient induction of ID1, compatible with the conditions we used (1 or 4 hours of acute stimulation). Other studies might have not identified ID1 as a TGF-beta target because of differences in the experimental conditions used. We have added a paragraph in the Discussion about the regulation and biological function of ID1.

7) They use AP activity as a readout for pluripotency. In the presence of SB, so lack of TGFb and related to more BMP signaling, the authors report more AP activity. AP is also a marker for osteogenic differentiation, which is induced by BMP. How are the authors able to discriminate between the two?

We thank the Referee for raising these points about a possible activation of BMP upon TGF-beta inhibition. In order to address it we first looked at phosphorylation of SMAD1/5/8, the BMP-induced Receptor-Smads, and observed no increase upon SB43 treatment (Supplementary Fig. 1c).

We have also analysed by RNAseq a set of genes, previously identified as BMP-induced in hESCs (Bernardo et al., 2011), and they were not induced by SB43. Such transcriptional analysis is shown in the Figure below:

SB43 treatment causes downregulation of pluripotency markers PRDM14 and NANOG and upregulation of the mesenchymal marker N-Cadherin. A panel of genes previously identified as induced by BMP in hPSCs (Bernardo et al., 2011) are then shown. They are either undetectable or their expression is not affected by SB43.

We conclude that in hESCs SB43 does not lead to BMP activation.

We should also point out that in presence of SB43 we always observe a reduction of AP activity (Fig. 2a, Supplementary Fig. 2c and Supplementary Fig. 3a), so a putative osteogenic differentiation should be reduced by SB43. Finally, we agree that AP staining is not only observed in pluripotent cells, therefore we performed the analysis of additional pluripotency markers (i.e. NANOG, PRDM14, OCT4/POU5F1) by qPCR, RNAseq and Immunostaining (Fig. 2c-f), which to our knowledge are not expressed by osteoblasts.

8) *In figure 3 they find genes related to mesenchymal transition and extracellular matrix organization, however this is normally related to TGFbeta signaling. In this study it is related to inhibition of TGFb signaling. Please explain.*

Please see our response to point 4. The effects on EMT markers and extracellular matrix we observed are coherent. The fact that in hPSCs, normal cells or transformed cells, TGF-beta signaling might activate or repress such gene program might be explained by the interplay with other signalling molecules, which will be investigated in the future.

9) *In figure 1A, please show the expression levels/localization of pSmad2, 3 and pSmad1 as inhibiting TGFbeta signaling often enhances BMP signaling, also important for pluripotency.*

As requested by Referee we now show the localization pSMAD3 by Immunostaining as well as pSMAD1/5/8 (Fig. 1a, Supplementary Fig. 1c). We observed the expected reduction of pSMAD3, while pSMAD1/5/8 was unchanged, in line with the lack of changes in BMP target genes (see Figure in point 7).

The expected changes in pSMAD3 were also confirmed by Western Blot (Supplementary Fig. 6c, d).

We have also performed analysis of the nuclear-cytoplasmic shuttling of SMAD2/3 upon SB43 treatment followed by stimulation (Supplementary Fig. 6b), finding rapid nuclear accumulation upon TGF-beta stimulation.

Concerning the role of BMP in pluripotent stem cells, we should mention that BMP has been used in murine ESCs, in combination with LIF, to maintain pluripotency (Ying et al., 2008). In such a context, BMP acts as a suppressor of Neuroectoderm fate via ID proteins, rather than a direct inducer of pluripotency. Conversely, in human ESCs, BMP is known as a potent inducer of Mesoderm (Bernardo et al., 2011).

10) *Why not feeder-free and activin? What receptors are expressed?*

As mentioned in points 2 and 3, we now show what receptors are expressed under feeder-free conditions (Supplementary Fig. 1f). Activin A and Nodal signal through the type I receptors ACVR1B/ALK4 and ACVR1C/ALK7 and the type II receptors ACVR2A/B, all of which are expressed in hPSCs. Indeed Activin and Nodal can be used for feeder-free culture, as demonstrated by Thompson and Vallier laboratories (Chen et al., 2011; Johansson and Wiles, 1995; Ludwig et al., 2006a, 2006b; Vallier, 2005). However, feeder-free conditions based on TGF-beta, such as E8 and mTeSR are more commonly used, therefore we decided to use those.

11) *Although significant, is the induction of Nanog and ZNF398 also relevant? Because fold of induction is shown, it is not clear what the levels are. Smad7, as a direct TGFb target, is much more induced. Is this a more relevant gene?*

We thank the Referee for pointing this out. Now we also show the endogenous expression levels of each target gene, making it easier to evaluate how relevant each induction is (Fig. 1f, bottom panel). For example, under feeder-free conditions it is clear that after 4 hours of TGF-beta stimulation SMAD7, NANOG, and ZNF398 all reach endogenous expression levels.

We should also point out that SMAD7 and LEFTY1, are known negative regulators and direct targets of TGF-beta/Activin, which we measured just to make sure that our treatment with ligands and inhibitors were successful.

Minor

Line 76 and to should be and the

Line 85 the down regulated gene

We have amended the text as suggested by the Referee.

Referee 2

In this manuscript, the authors reported the identification of the genes responsible for mediating the action of the TGF β signal in human pluripotent stem cells (hPSCs). The Activin/TGF β signal is essential for self-renewal of hPSCs in conventional culture condition but the molecular mechanism to mediate the action of this signal is largely unknown. The authors applied a sophisticated way of the analysis to identify the functional targets of the TGF β signal in hPSCs. As the result, they identified 4 genes encoding transcription factors (NANOG, MYC, KLF7 and ZNF398) that are regulated by the TGF β signal and sufficient to replace its function by their over-expression. Among them, the authors chose ZNF398 for further analysis. They found that ZNF398 co-occupy the target sites in the genome with SMAD3 and EP300. Moreover, they demonstrated that ZNF398 is essential for efficient reprogramming of the somatic cells to hPSCs by the OSKMNL factors. These results emphasize the role of ZNF398 to maintain and establish pluripotency.

ZNF398 is a mysterious factor since there is no previous report for its function in PSCs in any species. In somatic cells, it was recently reported that p52-ZER6 (encoded by ZNF398) interacts with TRP53 and modulates its stability (Huang et al, EBioMedicine, 2019: PMID 31521611).

We thank the Referee for mentioning this interesting work about the role of ZNF398/ZER6 in cancer cells. We were aware of the existence of two isoforms (p52 and p71), however in hESCs the longer isoform is the most expressed one. We had cloned and over-expressed both isoforms, but only the longer one (p71) had an effect on pluripotency (data not shown). It is quite interesting that only the short isoform acts a cell-cycle regulator, via the p21/p53 axis (Huang et al., 2019). We have added to the discussion a paragraph summarising all these findings.

According to HPA RNA-seq data in normal tissues, the expression of ZNF398 is quite homogeneous, suggesting its general function. Therefore, the inefficient reprogramming by the knockdown of ZNF398 could not solely due to the failure of the activation of pluripotency and MET-associated genes.

We have analysed the HPA RNA-seq data and found that ZNF398 is expressed at very low levels in several healthy tissues (Supplementary Fig. 7a). To put such results into perspective, we looked at the expression levels of the well-known pluripotency factor KLF4, which showed much more robust and widespread expression in healthy tissues. It is indeed common for pluripotency regulators to be expressed in differentiated tissues (e.g. the naive pluripotency factor Klf2 is also known as Lung KLF, given its function in lung development, while Sox2 is highly expressed in Neuroectoderm and in the adult cerebral cortex).

However, we agree on the fact that the inefficient reprogramming observed upon ZNF398 knockdown could be due to functions beyond the regulation of pluripotency and epithelial genes, such as proliferation.

We have experimentally tested this hypothesis by counting the number of cells at Day 6 of reprogramming and observed no difference between siControl- and siZNF398-treated samples. Moreover, markers of proliferation (CCNA2 and CDKN1A/p21) were not affected by ZNF398 knockdown (Supplementary Fig. 8e). We conclude that ZNF398 does not have an effect on proliferation of fibroblasts during reprogramming.

To further address concerns about a general function of ZNF398 we have also performed experiments in murine Epiblast Stem cells (EpiSCs) (Supplementary Fig. 4 and Supplementary Fig. 7d) and in human cancer cell lines (Supplementary Fig. 7b, c) which are discussed below.

The co-localization with SMAD3 may suggest its role to mediate the TGF β signal in any cell types, not specific to hPSCs. This point should be addressed carefully in the revised manuscript.

We have measured the expression of ZNF398 in a panel of cancer cell lines and we found that only two of them (HEK293T and HaCaT) showed levels comparable to those of human ESCs and iPSCs (Supplementary Fig. 7a). We performed loss of function studies in both cell lines, to test whether ZNF398 regulates TGF-beta signal in any cell type (Supplementary Fig. 7b).

Despite the robust downregulation of ZNF398, we observed no changes in the levels of TGF-beta targets SMAD7 and JUNB (Supplementary Fig. 7b). We have also performed transient over-expression of ZNF398 and again we could not observe any difference (Supplementary Fig. 7c).

Furthermore, we cloned the mouse ortholog of ZNF398 called Zfp398 and stably expressed it in two EpiSC lines (GOF18 and OEC2). We measured the levels of the TGF-beta targets Smad7, Lefty1, Lefty2 and Nodal (Supplementary Fig. 7d). The 4 target genes were all downregulated upon TGF-beta inhibition. However Zfp398 expression did not increase the expression of TGF-beta targets (Supplementary Fig. 7d), in stark contrast with what we observed in human PSCs (Fig. 4d, bottom panel, Fig. 4e-h).

Therefore we conclude that ZNF398 does not mediate TGF-beta signal in all cell types tested. Rather, its effect, like KLF7, appears specific for human pluripotent stem cells.

In addition, there are several points required revision for publication in Nat Commun.

1. In Fig. 2, the authors showed the impact of the overexpression of 4TFs they identified to maintain self-renewal in the absence of the TGF β signal. However, they look just short period of culture (for 5 days in the presence of SB43). How about their powers in longer culture period with passages in SB43?

As requested, we have expanded two hPSC lines (HES2 and KiPS) for ~15 days (4 passages) in SB43. Empty vector transfected cells rapidly lost expression of pluripotency markers NANOG, OCT4 and PRDM14 as well as AP activity (Supplementary figure S2b-d). Conversely, expression of the 4 TFs maintained AP activity and expression of pluripotency markers. As observed previously, each TF shows specificity for some targets. For example KLF7 induces robustly OCT4, while ZNF398 induces PRDM14.

In general, the magnitude of the effects observed after prolonged culture are comparable to those observed after 5 days in SB43, indicating stable maintenance of pluripotency by the 4 TFs.

2. Line 141: The authors stated the similarity between KLF7 and KLF2/4/5, but their distinct function to maintain pluripotency was demonstrated previously (Yamane et al, Development, 2018).

We apologise for our inaccurate statement, which we have amended in the revised text. What we meant is that several KLF factors, although acting through distinct mechanisms, have been already studied in pluripotent stem cells. In contrast, ZNF398 attracted our attention simply because it was completely novel, as also the Referee pointed out.

3. As mentioned above, ZNF398 could be a general co-factor of SMAD3 to support its function. The authors should address this possibility in appropriate somatic cell line that respond to the TGF β signal and express ZNF398.

We have addressed this point exactly performing the elegant experiments suggested by the Referee. We first identified two cell lines, HEK293T and HaCaT, which express ZNF398 at levels comparable to human ESCs and iPSCs (Supplementary Fig. 7a) and have been used to study TGF-beta signal. Then, we confirmed that they do respond to TGF-beta by looking at the induction of target genes upon TGF-beta stimulation. Finally we performed both knockdown and overexpression of ZNF398 and observed no significant difference in the induction of TGF-beta target genes (Supplementary Fig. 7b, c).

4. Line 270: *Is Dppa4 a human-specific pluripotency factor? Recent reports indicated that Dppa2/4 involve in the activation of 2C genes in mouse ESCs (PMID: 30948459, 30692203).*

We have amended the text and removed the incorrect statement. Dppa4 is indeed also expressed in murine ESCs.

5. *It will be nice if the authors show the function of the 4 TFs in mouse epiblast stem cells to replace the action of the TFGb signal.*

We have performed the over-expression of the TFs in two mouse EpiSCs lines (Supplementary Fig. 4a). We confirmed that TGF-beta inhibition caused loss of primed pluripotency markers, such as Nanog, Oct4, Ffg5 and Otx2. None of the 4 TFs were able to maintain pluripotency markers, with the exception of Otx2 being maintained by KLF7 (Supplementary Fig. 4b). We should mention that expression of the 4 TFs was comparable or superior to the one obtained in hPSCs and that the construct used are the same we used extensively for resetting of the same EpiSCs to naive pluripotency (Dunn et al., 2019). Therefore we believe that the lack of effects we observed is not due to technical issues.

Referee 3

Irene Zorzan and colleagues performed a global analysis of the transcriptional program controlled by TGF beta followed by an unbiased gain-of-function screening in multiple hPSC lines to identify factors mediating TGF beta activity. They found a new downstream mediator of TGF beta signal, ZNF398, which could induce genes associated both with pluripotency and epithelial character together with SMAD3 and the histone acetyltransferase EP300. This is an interesting work on a relatively poorly explored field, as it is human pluripotency, and it, thus, has potential relevant implications. In general, I liked the paper but before publication in Nature Communications there are several aspects that should be improved, in my opinion.

Major and minor problems:

1. In line 99 the authors mention 'extensive validation' but something relevant like a ChIP-qPCR panel is missing. Related, the plot in Figure 1e is not immediately intuitive.

We thank the Reviewer for the positive comments on our work.

We have performed ChIP-qPCR for SMAD2 and SMAD3 in two hPSC lines for the targets shown in Fig. 1f of the revised manuscript.

We have also made a new version of the balloon plot in Fig. 1e, whereby the same colours are used for all experiments and explained better in the legend. We think that it is important to show not only the fold-changes, but also the statistical significance of each comparison.

2. The evidence for co-localization and functional interaction of ZNF398 with SMAD3 and EP300 is not enough. The authors could do co-immunoprecipitation and/or ChIP-on-CHIP.

As requested by the Referee we have conducted co-immunoprecipitation and indeed observed the formation of a complex between SMAD3 and ZNF398 (Fig. 4c). Such results are in line with their co-localization (Fig. 4a, b) and with the increased expression of SMAD3 targets upon ZNF398 forced expression (Fig. 4d bottom panel, Fig. 4e-h).

3. Related to the previous item. In lines 205-207, the authors wrote "We extended our analysis to all 81 SMAD3 direct target genes (Fig. 1c) and observed that 28.4% of them were also co-bound by ZNF398 (enrichment of 3.67 fold over expected, p-value = 3.49e-12, Chi-squared test)". Please put this on a Figure to support the observation.

We have made new Fig. 4f panel to support the observation as requested

4. *The cut-offs in for example Figures 3a and 4f, aren't they too low?*

The thresholds used are $>+1$ or <-1 for the Log2Fold Change and >0.01 for the p-value. These are rather standard threshold, but we agree that the previous version of the Figs. 3a and 4h were too small or gave the wrong impression that the thresholds were too low. For this reason we have made new versions that are slightly larger and hopefully more clear.

5. ChIP-qPCR showing binding of ZNF398 to some target genes is desirable.

We have performed ChIP-qPCR in two hPSC lines (BG01V and H9) and confirmed binding on LIN28B, ESRP1 and LEFTY1 (Fig. 4d).

6. *The effect on epithelial genes in reprogramming after knockdown of ZNF398 is modest (e.g., Figures 5e and d). Since the authors reprogram the fibroblasts with redundant genes (according to their screen), wouldn't it be better to reprogram with less factors (e.g., the 4 Yamanaka factors) to see a more potent effect of the ZNF398 knockdown.*

As suggested by the Referee we have repeated the reprogramming experiments using only the 4 Yamanaka factors. We initially used 6 factors because we used a commercial kit, where the 6 were pre-mixed. So we first had to establish the protocol for *in vitro* transcription of each individual mRNA (details provided in the methods section). As expected, the efficiency of reprogramming with 4 Yamanaka factors is reduced compared to 6 factors (Fig. 5c).

In the case of reprogramming with the 4 Yamanaka factors the knockdown of ZNF398 almost completely abolished reprogramming (we obtained no colonies in 13 biological replicates out of 15, Fig 5c).

Moreover, the effects appeared modest because our previous qPCR analysis included both iPS colonies and cells that failed to reprogram. For this reason we have also performed immunostaining in the revised version, clearly showing that pluripotency and epithelial markers are specifically found in reprogrammed colonies (new Fig. 5d, f).

Related, the authors measure colony formation at day 14, which is not representative of reprogramming efficiency but rather the mesenchymal-to-epithelial transition, so please modify the statements.

We have performed immunostaining for NANOG and OCT4 to confirm that the colonies obtained expressed pluripotency markers. We scored our OSKM reprogramming experiments first based only on morphology (Figure 5c). Then, we performed immunostaining and scored again, obtaining highly similar results. In fact, all colonies identified by morphology were also NANOG and OCT4 positive.

7. Many references should be added to refer to the work of others. The figures should be mentioned in order to avoid confusion; Supplementary Figure 3c wasn't mentioned in the text. For Figure 4c, a different set of genes are mentioned in the text. Add error bars to Supplementary Figure 2b. Mention clearer how many times was each experiment repeated. Is Figure 3a first mentioned correctly in lines 134-136?

We have made all the suggested changes to improve the clarity and have added several references to other important studies in the field.

We have also clearly stated in the figure legends the number of independent experiments or biological replicates.

We will provide all such information as a Source Data table.

8. The methods need to be revised and explained in more detail. For example: why in lines 577 and 589 the authors use different tools? why the authors use a relatively old version of R (3.0) rather than an updated one? is the cut off in line 583 similar to the one used in the figures?

We have added details to the methods in order to improve clarity.

Concerning the use of different tools for the two indicated datasets, the reason is that we used two different sequencing platforms (Ion Proton and Illumina Nextseq 500) and for each one we used previously optimised analysis pipeline. All the tools used (e.g. bowtie, TopHat, edgeR and DESeq2) are commonly used for bulk RNA-seq analyses.

The version of R used was in fact 3.5.2, we indicated the version 3.0 as a mistake.

The cut offs indicated in lines 583 and 591 of the submitted version are exactly those used in the figures.

9. *Some parts of the text can be written clearer, for example when referring to TGF beta-like cytokines and receptors.*

We have re-written such paragraph and made a general effort to improve the clarity of the text.

REFERENCES

- Bernardo, A.S., Faial, T., Gardner, L., Niakan, K.K., Ortmann, D., Senner, C.E., Callery, E.M., Trotter, M.W., Hemberger, M., Smith, J.C., et al. (2011). BRACHYURY and CDX2 mediate BMP-induced differentiation of human and mouse pluripotent stem cells into embryonic and extraembryonic lineages. *Cell Stem Cell* 9, 144–155.
- Chen, G., Gulbranson, D.R., Hou, Z., Bolin, J.M., Ruotti, V., Probasco, M.D., Smuga-Otto, K., Howden, S.E., Diol, N.R., Propson, N.E., et al. (2011). Chemically defined conditions for human iPSC derivation and culture. *Nat. Methods* 8, 424–429.
- Chen, S., Choo, A., Chin, A., and Oh, S.K.W. (2006). TGF- β 2 allows pluripotent human embryonic stem cell proliferation on E6/E7 immortalized mouse embryonic fibroblasts. *J. Biotechnol.*
- Dennler, S., Itoh, S., Vivien, D., Dijke, P., and Gauthier, J. (1998). Direct binding of Smad3 and Smad4 TGF-PAI-1.pdf. *17*, 3091–3100.
- Dennler, S., Huet, S., and Gauthier, J.-M. (1999). A short amino-acid sequence in MH1 domain is responsible for functional differences between Smad2 and Smad3. *Oncogene* 18, 1643–1648.
- Dunn, S.-J., Li, M.A., Carbognin, E., Smith, A., and Martello, G. (2019). A common molecular logic determines embryonic stem cell self-renewal and reprogramming. *EMBO J.* 38, e100003.
- Huang, C., Wu, S., Li, W., Herkilini, A., Miyagishi, M., Zhao, H., and Kasim, V. (2019). Zinc-finger protein p52-ZER6 accelerates colorectal cancer cell proliferation and tumour progression through promoting p53 ubiquitination. *EBioMedicine* 48, 248–263.
- Johansson, B.M., and Wiles, M.V. (1995). Evidence for Involvement of Activin A and Bone Morphogenetic Protein 4 in Mammalian Mesoderm and Hematopoietic Development. *Mol. Cell. Biol.* 15, 141–151.
- Liang, Y.-Y., Brunicardi, F.C., and Lin, X. (2009). Smad3 mediates immediate early induction of Id1 by TGF-beta. *Cell Res.* 19, 140–148.
- Lucarelli, P., Schilling, M., Kreutz, C., Vlasov, A., Boehm, M.E., Iwamoto, N., Steiert, B., Lattermann, S., Wäsch, M., Stepath, M., et al. (2018). Resolving the Combinatorial Complexity of Smad Protein Complex Formation and Its Link to Gene Expression. *Cell Syst.* 6, 75-89.e11.
- Ludwig, T.E., Levenstein, M.E., Jones, J.M., Berggren, W.T., Mitchen, E.R., Frane, J.L., Crandall, L.J., Daigh, C.A., Conard, K.R., Piekarczyk, M.S., et al. (2006a). Derivation of human embryonic stem cells in defined conditions. *Nat. Biotechnol.* 24, 185–187.
- Ludwig, T.E., Bergendahl, V., Levenstein, M.E., Yu, J., Probasco, M.D., and Thomson, J.A. (2006b). Feeder-independent culture of human embryonic stem cells. *Nat. Methods* 3, 637–646.
- Ross, S., and Hill, C.S. (2008). How the Smads regulate transcription. *Int. J. Biochem. Cell Biol.* 40, 383–408.
- Senft, A.D., Costello, I., King, H.W., Mould, A.W., Bikoff, E.K., and Robertson, E.J. (2018). Combinatorial Smad2/3 Activities Downstream of Nodal Signaling Maintain Embryonic/Extra-Embryonic Cell Identities during Lineage Priming. *CellReports* 24, 1977–1985.e7.
- Singh, A.M., Reynolds, D., Cliff, T., Ohtsuka, S., Mattheyses, A.L., Sun, Y., Menendez, L., Kulik, M., and Dalton, S. (2012). Signaling Network Crosstalk in Human Pluripotent Cells: A Smad2/3-Regulated Switch that Controls the Balance between Self-Renewal and Differentiation. *Cell Stem Cell* 10, 312–

326.

Vallier, L. (2005). Activin/Nodal and FGF pathways cooperate to maintain pluripotency of human embryonic stem cells. *J. Cell Sci.* *118*, 4495–4509.

Ying, Q.L., Wray, J., Nichols, J., Battle-Morera, L., Doble, B., Woodgett, J., Cohen, P., and Smith, A. (2008). The ground state of embryonic stem cell self-renewal. *Nature* *453*, 519–523.

Zhang, Q., Yu, N., and Lee, C. (2014). Mysteries of TGF- β Paradox in Benign and Malignant Cells. *Front. Oncol.* *4*.

Reviewers' Comments:

Reviewer #1:

Remarks to the Author:

I want to congratulate the author with the resubmission which answered most of my questions.

Before this manuscript can be accepted, it is important to carefully read it again as quite some small errors are present in the text, e.g. (and not limited to):

Line 37-38: either refer to 1 type II binding to 1 type I, or if you want to be correct a heterotetrameric complex is formed. Sufficient recent reviews are available explaining the pathway.

Line 38: Activation of the receptorial complex: please refer to receptor complex

Line 41: Direct binding of the DNA is mediated by SMAD3: make the comparison with Smad2. Smad3 can bind DNA while Smad2 needs Smad4 to do so.

line 45: Thus: not clear why thus is written here.

line 53: activated by TGF-beta signal : This should be TGFb signaling and even better you might want to refer to TGFb/Smad3 signaling

Line 64: SB is an inhibitor of the ALK4/5/7 kinase. Not only TGFb1, but also TGFb2 and TGFb3 are inhibited. It blocks phosphorylation of Smad2 and Smad3 by the ALK4/5/7 kinase.

line 65: phosphorylation of Smad3 downstream of?

Line 71: TGF-beta is used, make sure you use the same abbreviation.

Line 73-74: I guess the transcription factor will bind the DNA and not the other way around.

Line 90: here the authors mention Activin A and TGFb1. Therefore it is important to mention that you are analyzing the TGFb pathway or TGFb/activin/smاد3 pathway, not to confuse the reader.

Line 95: bona fide TGF-beta: is the TGFb or also Activin

Line 218: accompanied by phosphorylation of SMAD3: first you get phosphorylation after which it will enter the nucleus. To really show that they are together and co-localize, PLA is necessary.

Suppl Figure 6 shows no nuclear translocation of Smad2/3, and 60 min of not as clear as the image in figure 1A for pSmad3, making it difficult to appreciate the data.

Reviewer #2:

Remarks to the Author:

In this revised manuscript, the authors addressed the points raised by the reviewers in proper way, and the manuscript is significantly improved. Now this reviewer is almost satisfied and mentions few points for further revision.

Line 48-49: The description is confusable. Among a large set of additional pluripotency factors identified in mouse ES cells, the majority is not expressed in human ES cells that are in primed pluripotent state, but most of them are expressed in naïve human PSCs. Throughout the manuscript, the authors use the term 'human PSCs' for human primed PSCs such as ES cells, but it could cover both naïve and prime PSCs in general.

Line 120: The authors stated that Myc might maintain pluripotency via other pluripotency factors. However, do the human primed PSCs maintain pluripotency with such low level expression of Oct4? The authors showed that overexpression of MYC maintain hESC-like morphology and AP activity but there is no direct evidence to verify their pluripotency. The authors should make proper statement reflecting these observations: maintain ES-like state etc.

Reviewer #3:

Remarks to the Author:

The authors have made a very significant effort in adding experiments and explanations to their

revised manuscript. In general, they have answered my concerns adequately and the resulting manuscript is much more solid, and interesting. The authors argue that immunofluorescence staining for OCT4 and NANOG shows that the colonies are already pluripotent at day 14. I disagree with that and the colonies are likely not there yet, or at least the authors don't prove it formally. I would suggest rephrasing this part a little bit, but I leave it to the discretion of the authors and ultimately it would not matter much because the manuscript is overall very good.

Response to Referees' comments

Referee #1

I want to congratulate the author with the resubmission which answered most of my questions.

We thank the Referee for appreciating our efforts.

Before this manuscript can be accepted, it is important to carefully read it again as quite some small errors are present in the text, e.g. (and not limited to):

Line 37-38: either refer to 1 type II binding to 1 type I, or if you want to be correct a heterotetrameric complex is formed. Sufficient recent reviews are available explaining the pathway.

Line 38: Activation of the receptorial complex: please refer to receptor complex

Line 41: Direct binding of the DNA is mediated by SMAD3: make the comparison with Smad2. Smad3 can bind DNA while Smad2 needs Smad4 to do so.

line 45: Thus: not clear why thus is written here.

line 53: activated by TGF-beta signal: This should be TGFβ signaling and even better you might want to refer to TGFβ/Smad3 signaling

Line 64: SB is an inhibitor of the ALK4/5/7 kinase. Not only TGFβ1, but also TGFβ2 and TGFβ3 are inhibited. It blocks phosphorylation of Smad2 and Smad3 by the ALK4/5/7 kinase.

line 65: phosphorylation of Smad3 downstream of?

Line 71: TGF-beta is used, make sure you use the same abbreviation.

Line 73-74: I guess the transcription factor will bind the DNA and not the other way around.

Line 90: here the authors mention Activin A and TGFβ1. Therefore, it is important to mention that you are analyzing the TGFβ pathway or TGFβ/activin/smad3 pathway, not to confuse the reader.

Line 95: bona fide TGF-beta: is the TGFβ or also Activin

Line 218: accompanied by phosphorylation of SMAD3: first you get phosphorylation after which it will enter the nucleus. To really show that they are together and co-localize, PLA is necessary. Suppl Figure 6 shows no nuclear translocation of Smad2/3, and 60 min of not as clear as the image in figure 1A for pSmad3, making it difficult to appreciate the data.

Thanks for the suggestions. We have edited the manuscript following all suggestions. We have made very clear when we were referring to TGF-beta or Activin A and deleted our statement about co-localization (Line 218). We have also asked a native English speaker (and scientist) to proofread the manuscript.

Referee #2

In this revised manuscript, the authors addressed the points raised by the reviewers in proper way, and the manuscript is significantly improved. Now this reviewer is almost satisfied and mentions few points for further revision.

Line 48-49: The description is confusable. Among a large set of additional pluripotency factors identified in mouse ES cells, the majority is not expressed in human ES cells that are in primed pluripotent state, but most of them are expressed in naïve human PSCs. Throughout the manuscript, the authors use the term 'human PSCs' for human primed PSCs such as ES cells, but it could cover both naïve and prime PSCs in general.

We are pleased to read the Referee #2 statement that the manuscript is significantly improved. We also think that his/her comments were very fair and constructive. We have clarified (see Introduction) that throughout the manuscript PSCs or iPSCs mean only primed/conventional human PSCs. We have also discussed that there are differences in what factors are expressed in human or murine PSCs, and certainly such differences could be explained by their naïve or primed pluripotent state. However, ZNF398 and KLF7 are specifically expressed and functional only in human PSCs and they are not expressed and functional in murine PSCs.

Line 120: The authors stated that Myc might maintain pluripotency via other pluripotency factors. However, do the human primed PSCs maintain pluripotency with such low level expression of Oct4? The authors showed that overexpression of MYC maintain hESC-like morphology and AP activity but there is no direct evidence to verify their pluripotency. The authors should make proper statement reflecting these observations: maintain ES-like state etc.

It is indeed true that we have not tested the pluripotency of MYC expressing cells. We have edited our conclusion accordingly.

Reviewer #3 (Remarks to the Author):

The authors have made a very significant effort in adding experiments and explanations to their revised manuscript. In general, they have answered my concerns adequately and the resulting manuscript is much more solid, and interesting. The authors argue that immunofluorescence staining for OCT4 and NANOG shows that the colonies are already pluripotent at day 14. I disagree with that and the colonies are likely not there yet, or at least the authors don't prove it formally. I would suggest rephrasing this part a little bit, but I leave it to the discretion of the authors and ultimately it would not matter much because the manuscript is overall very good.

Miguel A. Esteban

We thank Professor Esteban for appreciating our effort and for the constructive comments. We have edited the text as suggested, as we cannot conclude that the colonies were pluripotent at day 14, rather they simply express those two markers.